# The cytoplasmic phosphate level has a central regulatory role in the phosphate starvation response of *Caulobacter crescentus*

Maria Billini[1,2], Tamara Hoffmann[1,3], Juliane Kühn[1], Erhard Bremer[1,3] & Martin Thanbichler ®[1,2,3] ✉

In bacteria, the availability of environmental inorganic phosphate is typically sensed by the conserved PhoR-PhoB two-component signal transduction pathway, which uses the flux through the PstSCAB phosphate transporter as a readout of the extracellular phosphate level to control phosphate-responsive genes. While the sensing of environmental phosphate is well-investigated, the regulatory effects of cytoplasmic phosphate are unclear. Here, we disentangle the physiological and transcriptional responses of *Caulobacter crescentus* to changes in the environmental and cytoplasmic phosphate levels by uncoupling phosphate uptake from the activity of the PstSCAB system, using an additional, heterologously produced phosphate transporter. This approach reveals a two-pronged response of *C. crescentus* to phosphate limitation, in which PhoR-PhoB signaling mostly facilitates the utilization of alternative phosphate sources, whereas the cytoplasmic phosphate level controls the morphological and physiological adaptation of cells to growth under global phosphate limitation. These findings open the door to a comprehensive understanding of phosphate signaling in bacteria.

Phosphorus is the fifth most abundant element in living organisms and has a critical role in a variety of essential cellular processes, including DNA and RNA synthesis, membrane biogenesis, cellular energy storage, protein modification and signal transduction. In biological systems, it is predominantly assimilated in the form of inorganic phosphate ($P_i$). Therefore, monitoring the extracellular and maintaining the appropriate intracellular phosphate concentration are vital tasks that require dedicated and efficient systems for phosphate sensing, uptake and assimilation[1]. In many environments, such as soil and oligotrophic aquatic habitats, the productivity of ecosystems is limited by an insufficient supply of phosphorus[2,3]. As a response to phosphate limitation, bacteria attempt to make use of existing inorganic or organic phosphate sources by redirecting intracellular phosphate metabolism and storage or by inducing the synthesis of phosphate transport systems and enzymes converting organic to inorganic phosphate. These processes involve large-scale changes in gene expression. Many phosphate-regulated genes are part of the so-called Phosphate (Pho) regulon, which was first characterized in *Escherichia coli* and subsequently in other bacteria[4–6]. The control of the Pho regulon is mediated by a conserved two-component system comprising the membrane-associated bifunctional histidine kinase PhoR and its cognate DNA-binding response regulator PhoB (names apply to *E. coli* and may differ in other species)[7–9].

Low concentrations of inorganic phosphate in the growth medium stimulate the kinase activity of PhoR and thus lead to the phosphorylation of the receiver domain of PhoB[1,10]. Phosphorylated PhoB then binds stably to specific DNA sequences (PHO boxes) that are located upstream of target genes and activates their expression by recruiting RNA polymerase[11–13]. In this way it controls, directly and indirectly, the expression of numerous genes involved in the adaptation of cells to phosphate limitation[1]. Although PhoR is membrane-associated, it lacks a periplasmic sensing domain and is thus unable to directly sense the extracellular phosphate concentration[14,15] (Fig. 1A).

A key player in the sensing of extracellular phosphate is the PstSCAB high-affinity $P_i$ transport system, a member of the ATP-binding cassette (ABC) transporter superfamily[16]. Deletions in the *pst* genes result in constitutive activation of the Pho regulon, regardless of the environmental phosphate levels[17]. Previous studies have suggested a model in which PhoR senses different conformational states that PstSCAB adopts during its transport cycle[18] (Fig. 1A). In this model, the phosphate-loaded state of the transporter signals a phosphate-replete environment, keeping PhoR in the

[1]Department of Biology, University of Marburg, 35043 Marburg, Germany. [2]Max Planck Institute for Terrestrial Microbiology, 35043 Marburg, Germany. [3]Center for Synthetic Microbiology, 35043 Marburg, Germany. ✉e-mail: thanbichler@uni-marburg.de

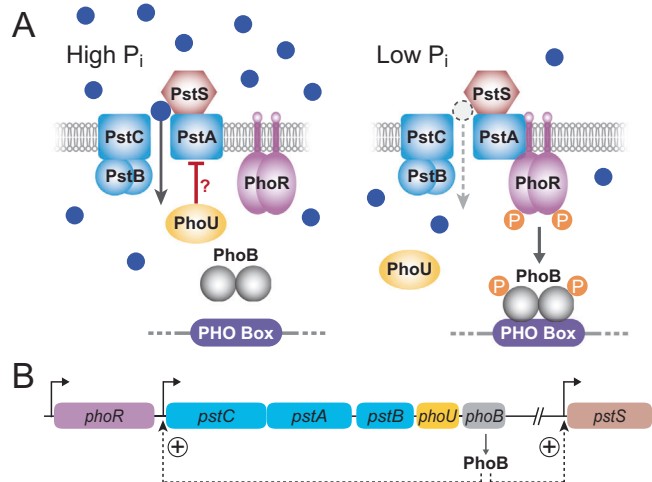

**Fig. 1 | Mechanism of environmental phosphate sensing by the PstSCAB-PhoRB pathway. A** Control of the activation state of the response regulator PhoB through the transport activity of the PstSCAB high-affinity phosphate transport system, as described for *C. crescentus*[22,25]. At high environmental phosphate concentrations, the PstSCAB system is predominantly in a substrate-bound state, which prevents the phosphorylation of the histidine kinase PhoR and thus the phosphorylation of the regulator PhoB. At low phosphate concentrations, the vacant form of the PstSCAB system activates PhoR, which then phosphorylates PhoB, thereby promoting its interaction with specific binding sites (PHO boxes) in target promoter regions. PhoU is a regulatory protein that may control the transport activity of the PstSCAB system[21,22]. **B** Organization of the *phoR*, *pstCAB-phoUB* and *pstS* genes and activating effect of phosphorylated PhoB on *pstSCAB-phoUB* and *pstS* expression.

phosphatase mode. By contrast, an empty transporter signals a phosphate-limited environment, switching PhoR to the kinase mode and activating PhoB. In *E. coli*, the interplay between PstSCAB and PhoR appears to be mediated by the cytoplasmic metal-binding protein PhoU, because inactivation of PhoU leads to constitutive activation of the PhoR kinase activity irrespective of the activity state of PstSCAB[19,20]. However, this function of PhoU may not be conserved in other species[21,22].

Phosphate limitation does not only change the physiology of cells, but it can also lead to specific morphological transformations. A prominent example is the alphaproteobacterium *Caulobacter crescentus*, a species characterized by a cellular extension, called the stalk, which carries an adhesive holdfast at its tip and serves as an attachment structure[23]. Phosphate limitation, but not other kinds of nutrient stress, induces a considerable (up to ~20-fold) increase in the length of the stalk, accompanied by a moderate elongation and thinning of the cell body[24–27]. Moreover, it leads to an arrest of DNA replication and cell division, establishing a non-proliferative state in which cells can exist for an extended period of time[24]. Similar to other bacteria, *C. crescentus* has a *pstSCAB* operon, whose expression is strongly upregulated by PhoB under phosphate limitation[22,25] (Fig. 1B). Based on genome analysis and $P_i$-uptake assays, it does not appear to possess any other high-affinity phosphate uptake system[25], such as homologs of the *E. coli* PitA or PitB proteins[28], although it contains transporters for alternative phosphorus sources such as phosphonates. The deletion of any of the *pst* genes mimics a phosphate-poor environment and activates the Pho regulon via the PhoR-PhoB pathway, supporting the notion that the PstSCAB system is the main entry point for phosphate in *C. crescentus* cells[22,25,28]. In addition, the mutant cells show the same long-stalk morphology as phosphate-starved cells, even when grown in phosphate-replete conditions[22,25]. By contrast, cells bearing a deletion of *phoB* fail to efficiently elongate their stalks upon phosphate starvation. Moreover, both *pst* and *phoB* mutants show reduced growth rates, even when cultivated in phosphate-replete medium, likely due to inefficient phosphate assimilation. Based on these findings, the characteristic long-stalk phenotype of *C. crescentus pst* mutants was attributed to the activation of PhoB[25], and stalk

length has since then served as a standard marker for the phosphorylation state of PhoB and, hence, the availability of environmental phosphate in this species. A combination of ChIP-seq and microarray analysis of phosphate-starved wild-type cells as well as *phoB* and *pstS* mutant cells enabled the identification of the direct *C. crescentus* Pho regulon[22]. A comparison with the Pho regulon of *E. coli*[6,29] revealed considerable differences in the nature of the genes regulated by PhoB, indicating that the composition of the Pho regulon may vary with the physiology of a species and the environmental niche it inhabits.

Overall, the ability to sense and respond to extracellular phosphate through the PstSCAB-PhoR-PhoB pathway appears to be conserved across species. Notably, there is evidence that this pathway may be additionally modulated by the intracellular phosphate level in at least two bacterial species, *E. coli* and *Salmonella enterica*[30,31]. Moreover, in *C. crescentus* and *Sinorhizobium meliloti*, PhoU was shown to sense, directly or indirectly, cytoplasmic phosphate and prevent its accumulation to toxic levels, likely by directly inhibiting the activity of the PstSCAB transport system[21,22] (Fig. 1A). Despite these examples, the range of mechanisms underlying intracellular phosphate sensing and their degree of conservation remain unclear. In many organisms, the presence of multiple, partly redundant $P_i$ transporters as well as the cross-activation of PhoB by histidine kinases other than PhoR (Fisher et al.[32]) make it difficult to address this issue. *C. crescentus*, by contrast, lacks alternative phosphate transporters, and its PhoB homolog appears to be controlled exclusively through the PstSCAB-PhoR pathway[33] (Fig. 1B), which makes this species an excellent model to study the principles of phosphate regulation.

In this study, we reinvestigated the response of *C. crescentus* to phosphate limitation by disentangling the sensing of the extracellular and cytoplasmic phosphate levels. To this end, we generated strains that produced the PitA transporter of *E. coli* and were thus able to take up $P_i$ independently of the PstSCAB system or its activator PhoB. In phosphate-replete medium, the presence of PitA abolished both the growth defect and the characteristic long-stalk phenotype of a Δ*pstS* mutant, while known components of the Pho regulon remained upregulated. This finding indicates that the core PhoR-PhoB signaling pathway only senses the levels of extracellular phosphate, dependent on the transport activity of the PstSCAB system, but is blind to changes in the cytoplasmic phosphate pool. Moreover, it demonstrates that the morphological adaptation of *C. crescentus* to low-phosphate environments is largely dependent on a thus-far unknown pathway that senses the availability of $P_i$ within the cell. Whole transcriptome analysis of various mutant strains in the absence and presence of PitA then enabled us to determine the contributions of the PstSCAB-PhoRB pathway and the cytoplasmic phosphate sensing pathway to the global transcriptional response of *C. crescentus* to phosphate limitation. Collectively, our study reveals complementary roles of extracellular and intracellular phosphate sensing in the control of cell shape and cell physiology in *C. crescentus*. In addition, it provides a straightforward approach to discriminate between these two different sensing strategies that is readily applicable to other model systems.

## Results

### Deletion of *phoB* reduces the fitness of *C. crescentus* cells under phosphate starvation

Upon cultivation in phosphate-limited media, *C. crescentus* cells display slow growth and develop distinct morphological features such as long stalks and elongated cell bodies[23]. Previous work has shown that this behavior is also observed in phosphate-rich medium when cells lack the periplasmic phosphate-binding protein PstS and, thus, are unable to take up phosphate from the environment at physiologically adequate rates[25]. A slow-growth phenotype was also observed for PhoB-deficient cells[25], since PhoB is required for proper *pstSCAB* expression and its absence thus also leads to a shortage of intracellular phosphate[22]. Interestingly, however, the *phoB* mutant no longer forms highly elongated stalks in response to phosphate depletion, which gave rise to the notion that the processes leading to stalk elongation may be regulated directly by the PstSCAB-PhoRB pathway[25]. To

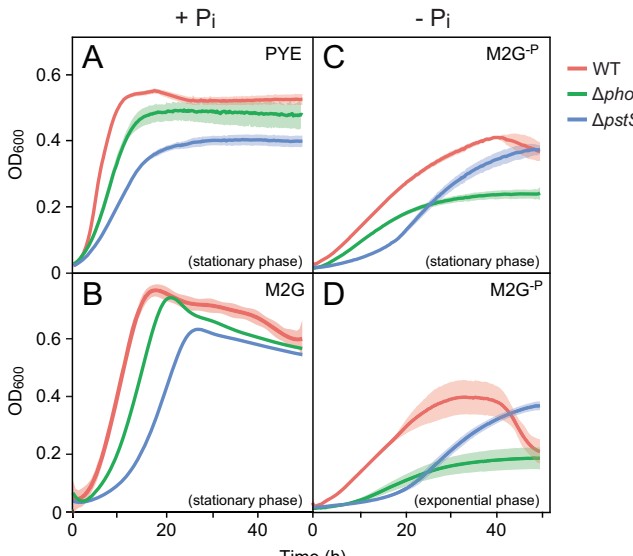

**Fig. 2 | Growth behavior of *C. crescentus* strains in phosphate-rich and phosphate-free media.** Shown are growth curves of wild-type (CB15N), Δ*phoB* (JK2) and Δ*pstS* (JK158) cells in phosphate-rich (+P$_i$) (**A**) complex (PYE) or (**B**) minimal (M2G) medium or in (**C**, **D**) phosphate-free (−P$_i$) minimal medium (M2G$^{-P}$). The cultures were inoculated (1:20) with cells cultivated to the exponential or stationary growth phase in PYE medium, as indicated in the graphs. Lines represent the average of three independent experiments. Shades indicate the standard deviation.

further investigate this possibility and set the stage for a subsequent global analysis of phosphate regulation in *C. crescentus*, we first re-analyzed the morphology and growth behavior of strains carrying in-frame deletions in the *pstS* and *phoB* genes. The results confirmed the abovementioned previous findings, which were obtained with insertion mutants[25]. Specifically, they showed that, in phosphate-replete complex medium (PYE; containing ~0.2 mM phosphate), the Δ*phoB* mutant only showed moderate defects in cell growth and morphology, whereas both of these traits were strongly affected in the case of the Δ*pstS* mutant (Fig. 2A and Supplementary Fig. 1). Similar behaviors were observed in minimal medium containing an elevated concentration (10 mM) of inorganic phosphate (Fig. 2B). Considering that the Δ*phoB* mutant still shows basal *pstSCAB* expression, whereas the lack of PstS completely abolishes phosphate uptake, the severity of the growth phenotypes thus scales with the decrease in the phosphate transport capacity.

Next, we analyzed the growth behavior of cells under phosphate starvation, induced by dilution (1:20) from phosphate-replete (PYE) medium into phosphate-free (M2G$^{-P}$) minimal medium. Wild-type cells continued to grow, albeit at a reduced rate (Fig. 2C, D). Even slower growth was observed when the cells were washed before transfer into M2G$^{-P}$ medium, and no further increase in cell density was measured after a second round of dilution into the same medium (Supplementary Fig. 2A). This result indicates that the growth observed in phosphate-free conditions can be attributed to residual phosphate carried over with the inoculum and intracellular phosphate storage compounds. Importantly, irrespective of their previous treatment, wild-type cells resumed growth with the same timing and to the same extent when transferred into rich medium (Supplementary Fig. 2A). Thus, the degree of phosphate starvation did not have any obvious effect on their viability. Unlike the wild-type strain, the Δ*pstS* mutant showed an extended lag phase after its transfer into phosphate-free (M2G$^{-P}$) medium, likely due to its inability to take up residual phosphate, but then reached a comparable growth rate and final cell density (Fig. 2C, D). The Δ*phoB* mutant, by contrast, did not show a conspicuous lag phase under these conditions, but it ceased to grow after a relatively short period of time (Fig. 2C, D). This effect was even more pronounced when the cells were

washed before transfer into phosphate-free medium (Supplementary Fig. 2A). Moreover, Δ*phoB* cells took considerably longer to recover from phosphate starvation than wild-type cells, with their lag phase depending on the extent of phosphate limitation (dilution or washing) before the starvation phase (Supplementary Fig. 2A). To clarify the reason for this phenomenon, we quantified the number of wild-type and Δ*phoB* cells after prolonged phosphate deprivation. Notably, when equal numbers of wild-type and Δ*phoB* cells were cultivated in phosphate-free medium for three days (72 h), the Δ*phoB* mutant reached a considerably lower final cell count than the wild-type strain (Supplementary Fig. 2B). Its reduced ability to grow and elongate the stalk in low-phosphate media may therefore be explained, at least in part, by a general growth and fitness defect rather than a specific defect in stalk biogenesis caused by the disruption of PhoB signaling. Collectively, these phenotypic analyses demonstrate that cell growth is markedly influenced by the rate of phosphate uptake in phosphate-replete environments and strongly dependent on the activation of the PhoB regulon in phosphate-limiting conditions.

## Production of PitA induces wild-type growth and morphology in Δ*pstS* and Δ*phoB* mutants

So far, it has been difficult to differentiate between the response of *C. crescentus* to extracellular phosphate, mediated by the PstSCAB-PhoRB pathway, and the regulatory effects exerted by cytoplasmic phosphate, because mutations impairing PhoB activity also reduce *pstSCAB* expression[22] and, thus, the transport of phosphate into the cell[25]. We therefore aimed to uncouple PhoB signaling from phosphate uptake by producing the phosphate transporter PitA from *E. coli* in *C. crescentus* cells. To determine whether PitA was functional in this heterologous system, we expressed its gene under the control of a xylose-inducible promoter in the Δ*pstS* background. Subsequently, we analyzed the resulting strain for its ability to take up radiolabeled phosphate and compared the results to those obtained for a wild-type and a Δ*pstS* control strain. To ensure the upregulation of the PstSCAB transport system in the wild-type strain, all three strains were starved for phosphate prior to the start of the measurements. The transport kinetics observed indicate that PitA mediated efficient and high-affinity phosphate uptake, with a $v_{max}$ of 4.7 nmol/min per OD$_{600}$ unit and a $K_M$ of 1.9 µM (Fig. 3). Very similar values were obtained for wild-type cells with a fully induced PstSCAB system, whereas only low rates of phosphate uptake were measured for a Δ*pstS* mutant. These findings indicated that PitA should be able to functionally replace the endogenous PstSCAB system.

After having established PitA as an alternative phosphate uptake system, we went on to resolve the responses of *C. crescentus* to changes in the external and internal phosphate levels. To this end, we compared the morphology and growth behavior of wild-type, Δ*pstS* and Δ*phoB* cells in the absence and presence of heterologously produced PitA. Surprisingly, in phosphate-replete medium, the presence of PitA largely abolished the mutant phenotypes of the Δ*pstS* and Δ*phoB* strains, yielding cells with short stalks and wild-type-like cell bodies (Fig. 4 and Supplementary Fig. 3) whose growth behavior was indistinguishable from that of the wild-type strain (Fig. 5). These findings indicate that the slow-growth phenotype of the two mutant strains is caused by insufficient phosphate uptake, resulting from impaired PstSCAB function (Δ*pstS*) or production (Δ*phoB*). Moreover, they demonstrate that stalk elongation is primarily induced by depletion of the cytoplasmic phosphate pool, independently of PstSCAB-dependent PhoR-PhoB signaling. Thus, *C. crescentus* does not only sense the availability of extracellular phosphate, using the activity of the PstSCAB transporter as a read-out, but also possesses thus-far unidentified sensory mechanisms that responds to the cytoplasmic phosphate level.

Notably, when diluted from phosphate-replete into phosphate-free medium, wild-type cells producing PitA showed higher growth rates and cell densities than cells without PitA. Moreover, Δ*pstS* cells producing PitA no longer displayed a lag phase and exhibited a growth behavior similar to that of wild-type cells without PitA. The presence of PitA thus appears to promote growth in low-phosphate conditions, probably by increasing the ability of cells to scavenge the residual phosphate introduced with the

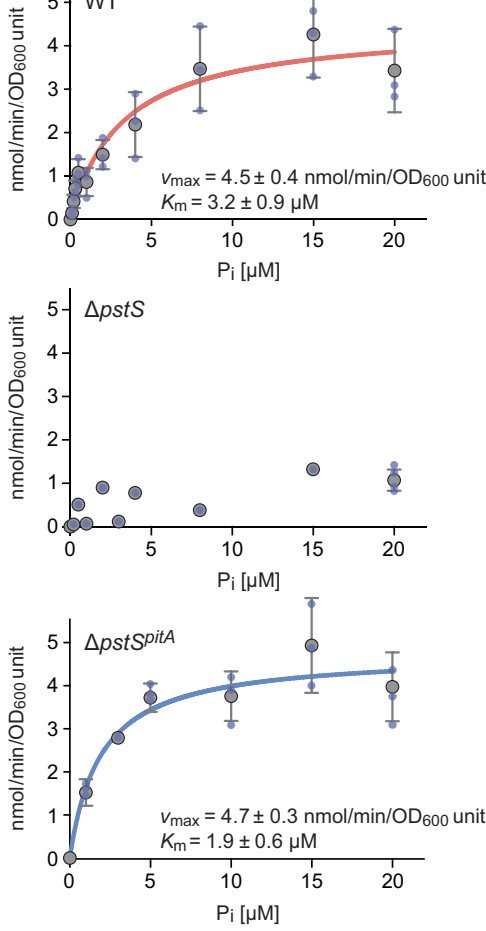

**Fig. 3 | Kinetics of phosphate uptake into different *C. crescentus* strains.** Shown are the kinetics of phosphate uptake into wild-type (CB15N), Δ*pstS* (JK158) and Δ*pstS*[pitA] (MAB215) cells. Cells were pre-grown in PYE medium with xylose, washed and incubated for 6 h in M2G[-P] medium with xylose prior to the measurements. To determine the rates of phosphate uptake over time, cells were incubated with the indicated concentrations of radioactively labeled phosphate. Subsequently, samples of the cultures were harvested by filtration at regular time intervals, and the amount of phosphate taken up was determined by scintillation counting. The uptake rates determined at the different phosphate concentrations were fitted to a Michaelis-Menten equation. The resulting $v_{max}$ and $K_M$ values are indicated in the graphs. Data represent the mean of three independent measurements (± SD), unless indicated otherwise.

inoculum and to accumulate phosphate storage compounds, such as polyphosphate[34], during cultivation in phosphate-replete medium, which then facilitate growth during phosphate deprivation. Importantly, however, in phosphate-limited conditions, the production of PitA did not alleviate the growth defect of the Δ*phoB* mutant, which underscores the importance of PhoB for cellular fitness during phosphate deprivation (see also Supplementary Fig. 2).

## PhoB signaling is not affected by the cytoplasmic phosphate level

It was conceivable that the response of *C. crescentus* to changes in the cytoplasmic phosphate level also involved PhoB, albeit in a manner independent of the PstSCAB system. To clarify this point, we analyzed the activation state of PhoB in wild-type and Δ*pstS* cells in the absence or presence of PitA, using the activities of the previously characterized PhoB-dependent *pstS* and *pstC* promoters[22] as a readout. For this purpose, each of the two promoters was fused to a *lacZ* reporter gene, and the expression levels of the fusion constructs were quantified using β-galactosidase assays. In addition, we measured the activities of the *ppk1* and *CCNA_01606*

promoters, which we had previously found to be upregulated during phosphate starvation although they were not members of the PhoB regulon[22]. As expected, when assayed in phosphate-replete medium, all four promoters showed only basal activity in the wild-type background, whereas they were highly upregulated in Δ*pstS* cells (Fig. 6A). Importantly, the activities of the two PhoB-dependent promoters did not drop in the presence of PitA but rather increased slightly. The two PhoB-independent promoters, by contrast, almost returned to their basal activity levels when PitA was produced and the cytoplasmic phosphate pool was restored (Fig. 6B). Taken together, these findings show that the activation state of PhoB is largely unaffected by the cytoplasmic phosphate level and regulated by the availability of extracellular phosphate as sensed by the PstSCAB-PhoR system. Moreover, they confirm that the replenishment of the cytoplasmic phosphate pool by PitA was sufficient to restore wild-type morphology and growth in the Δ*pstS* background, even though the PhoB regulon remained upregulated under this condition.

## Identification of genes specifically regulated by the cytoplasmic phosphate level

The results described above pointed to the existence of two distinct regulons responding to a shortage of environmental or cytoplasmic phosphate, respectively. To characterize these regulons in more detail and clarify to what extent they overlapped, we performed RNA-seq-based transcriptome analyses of wild-type, Δ*pstS* and Δ*phoB* cells in the absence (WT, Δ*pstS*, Δ*phoB*) or presence (WT[pitA], Δ*pstS*[pitA], Δ*phoB*[pitA]) of PitA after cultivation in phosphate-replete medium. The strains analyzed had distinct characteristics with respect to the cytoplasmic phosphate level and the activation state of PhoB (Table 1). A first analysis of the gene expression profiles showed that samples with similar properties formed distinct clusters in a multidimensional scaling plot, which qualified the data for further analysis (Supplementary Fig. 4A). Moreover, we observed that known PhoB-regulated genes, including the *pstCAB-phoUB* cluster and *pstS*, remained upregulated in Δ*pstS*[pitA] cells, confirming that PhoB remained activated in this condition (Supplementary Fig. 4B).

Having verified the validity of the data, we went on to identify genes whose expression was regulated specifically by changes in the cytoplasmic phosphate pool. To this end, we first compared the data obtained for Δ*pstS* cells, which exhibit a low cytoplasmic phosphate level against those obtained for WT, WT[pitA] and Δ*pstS*[pitA] cells, which all exhibit high cytoplasmic phosphate levels. All of these strains contain an intact PhoB protein, present in either a high (Δ*pstS*, Δ*pstS*[pitA]) or basal (WT, WT[pitA]) activity state. Depending on their genetic background, strains producing PitA may accumulate cytoplasmic phosphate to varying degrees and to levels different from those in the wild-type cells, a factor that should be considered in the interpretation of the results below. The comparison of Δ*pstS* with wild-type cells identified genes that were regulated under phosphate starvation, including those whose expression was controlled by extracellular phosphate through the PstSCAB-PhoRB pathway. By additionally comparing Δ*pstS* with WT[pitA] cells, which possess two phosphate uptake systems and may thus over-accumulate cytoplasmic phosphate, we considered potential effects of PitA on phosphate-dependent signaling. At last, the subset of genes specifically responding to changes in the cytoplasmic phosphate pool was identified by comparing Δ*pstS* with Δ*pstS*[pitA] cells. Importantly, this analysis also subtracted the direct contribution of PhoB activation from the global phosphate starvation response. Details of the significantly regulated genes obtained for each case are given in Supplementary Fig. 5 and Supplementary Data 1. The three comparisons revealed a set of 251 genes that were robustly regulated by changes in the cytoplasmic phosphate levels (Fig. 7A and Supplementary Data 1). However, since all strains used in this analysis still contained PhoB, either in a high or basal activity state, this set may include genes that respond, potentially cooperatively, to both the cytoplasmic phosphate level and PhoB.

To further refine our analysis, we performed a second set of comparisons, using the Δ*phoB* mutant as a reference. Specifically, the transcriptional profile of Δ*phoB* cells, which experience moderate phosphate deprivation,

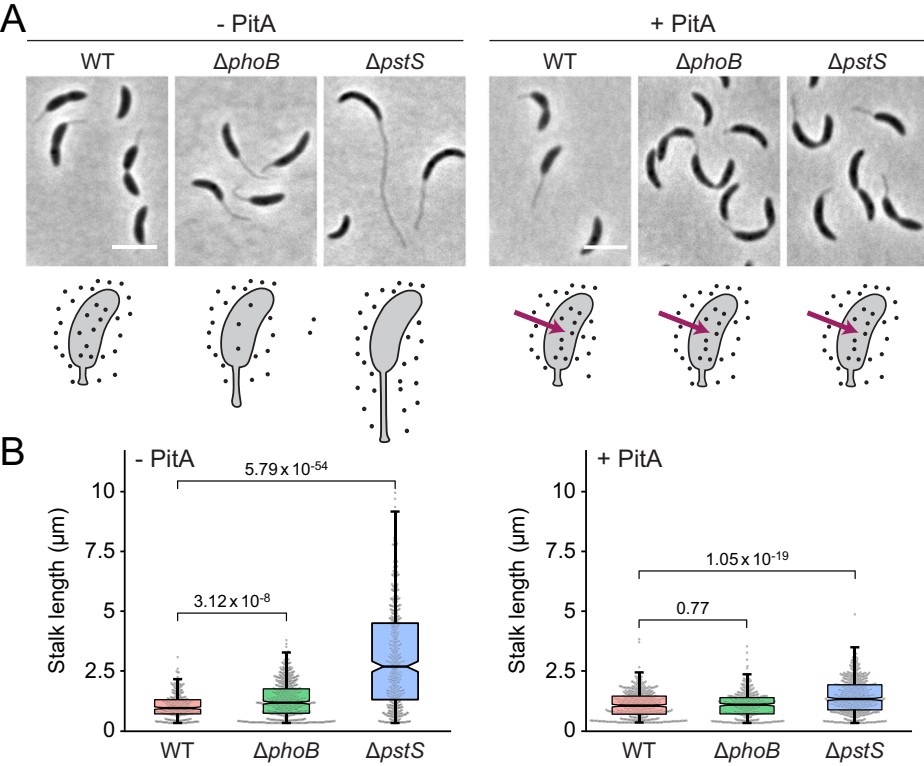

**Fig. 4 | Abolishment of the slow-growth and stalk elongation phenotypes of *C. crescentus* Δ*phoB* and Δ*pstS* mutants upon heterologous expression of *E. coli pitA*. A** Phase contrast images of wild-type (MAB257), Δ*phoB* (MAB258) and Δ*pstS* (MAB259) cells carrying the *pitA* gene under the control of a xylose-inducible promoter, grown to mid-exponential phase in PYE medium in the absence (− PitA) or presence (+ PitA) of inducer (scale bar: 3 µm). The schematics at the bottom illustrate the levels of phosphate (black dots) in the cytoplasm of the respective strains. Red arrows indicate PitA transport activity. **B** Combined beeswarm and box plots representing the distribution of stalk lengths in cultures of the strains shown in (**A**). The boxes give the interquartile range, the notches indicate the median values, and the whiskers extend to the 5th and 95th percentile. Number of cells measured: WT (313), Δ*phoB* (622), Δ*pstS* (519) in the absence of xylose (- PitA) and WT (485), Δ*phoB* (372), Δ*pstS* (567) in the presence of xylose (+ PitA). Numbers indicate the statistical significance (*p* values) of differences between strains (unpaired, two-tailed t-test).

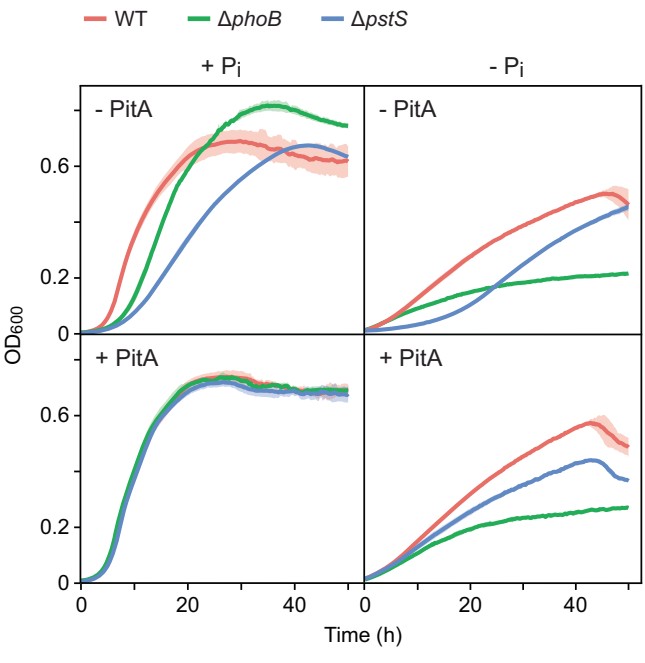

**Fig. 5 | Growth behavior of *C. crescentus* strains upon heterologous expression of *E. coli pitA*.** Shown are the growth curves of wild-type, Δ*phoB* and Δ*pstS* cells lacking *pitA* (CB15N, JK2, JK158) (- PitA) or carrying *pitA* under the control of a xylose-inducible promoter (MAB257, MAB258, MAB259) (+PitA). The cultures were grown in PYE medium (+P$_i$) or pre-grown in PYE medium and then diluted (1:20) into M2G$^{−P}$ medium prior to further incubation (−P$_i$). For strains carrying *pitA*, media were supplemented with xylose. Lines represent the average of two independent experiments. The edges of the shades indicate the values obtained in the two measurements.

was compared to that of WT, WT$^{pitA}$ and Δ*phoB*$^{pitA}$ cells, which all exhibit high, but different cytoplasmic phosphate levels. The comparison of Δ*phoB* with WT and WT$^{pitA}$ cells identified genes differentially regulated by changes in phosphate availability as well as PhoB activity (absent vs. basal), and it

considered potential effects induced by the heterologous production of PitA. The comparison of Δ*phoB* with Δ*phoB*$^{pitA}$ cells, on the other hand, revealed genes that responded to fluctuations in the cytoplasmic phosphate levels irrespectively of the absence of PhoB. Taken together, these analyses defined a set of 236 genes that showed a robust, PhoB-independent regulation by cytoplasmic phosphate (Fig. 7B, Supplementary Data 1 and Supplementary Fig. 5).

By combining the two gene sets described above, one focusing on severely phosphate-starved Δ*pstS* cells and one on moderately phosphate-starved Δ*phoB* cells as a reference, we obtained an intersection of 88 genes whose expression changed robustly in response to fluctuations in the cytoplasmic phosphate level, irrespective of the activity state or the absence/presence of PhoB (Fig. 7C and Supplementary Data 1). This analysis also yielded two additional gene sets, one comprising 163 genes that responded only to a severe shortage of cytoplasmic phosphate (with PhoB present) and another one comprising 148 genes that only changed under moderate cytoplasmic phosphate deprivation (with PhoB absent). Since the strains used to define these two additional sets either all contained or all lacked PhoB, we cannot exclude the possibility that some of these genes may require, directly or indirectly, the presence or absence of PhoB to respond to changes in the cytoplasmic phosphate level.

A clustering analysis that compared the expression levels of the final three gene sets further illustrated the marked changes in gene expression induced by the induction of PitA in the Δ*phoB* and Δ*pstS* mutants (Fig. 7D and Supplementary Figs. 6 and 7). Moreover, it identified distinct groups of genes whose expression levels either increased or decreased differentially in the different mutant backgrounds, depending on the cytoplasmic phosphate level. Together, these results reveal that *C. crescentus* possesses pathways to specifically sense and respond to the availability of cytoplasmic phosphate that act independently of PhoB signaling. Moreover, they point to complementary roles of the cytoplasmic and the PhoB-dependent extracellular phosphate sensing pathways in the overall response to phosphate deprivation.

**Functions of genes regulated by the cytoplasmic phosphate level**
To further validate the results obtained, we compared the lists of cytoplasmic phosphate-regulated genes with the previously identified PhoB regulon[22].

**Fig. 6 | Effect of PitA production on the activity of different phosphate-responsive promoters.** Wild-type (CB15N), WT$^{pitA}$ (MAB213), Δ$pstS$ (JK158) and Δ$pstS^{pitA}$ (MAB215) cells were transformed with reporter plasmids that carried fusions of (**A**) the PhoB-dependent $pstS$ (pMAB34) or $pstC$ (pMAB104) promoters or (**B**) the PhoB-independent $ppk1$ (pMAB106) or $CCNA\_01606$ (pJR16) promoters to the $lacZ$ gene. Transformants were grown in PYE medium supplemented with xylose and subjected to β-galactosidase activity assays. Data represent the mean (± SD) of three independent measurements.

## A PhoB-dependent promoters

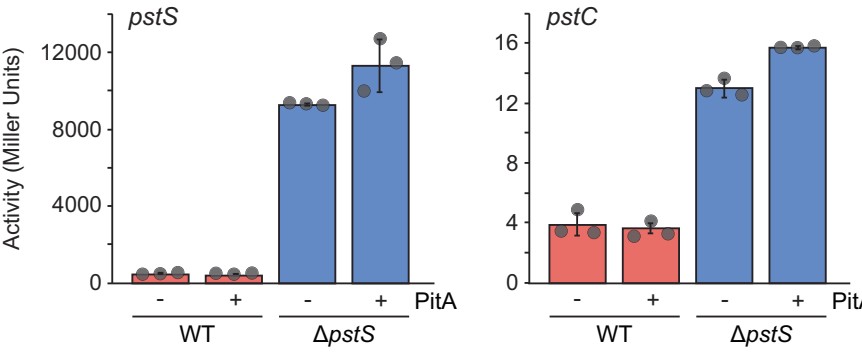

## B PhoB-independent promoters

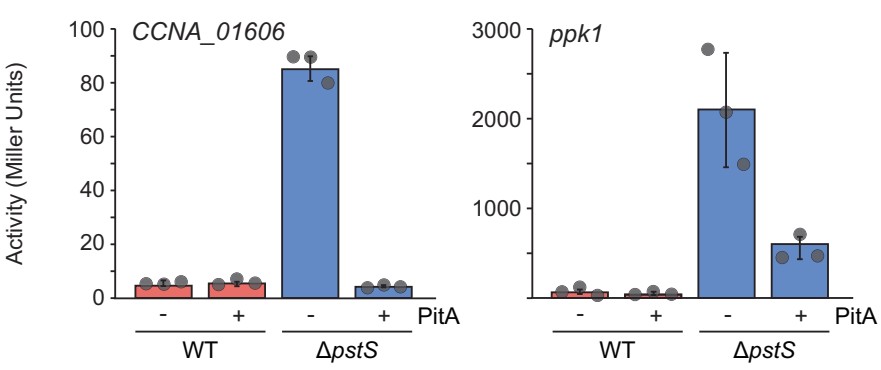

There was no overlap between the PhoB regulon and the core cytoplasmic phosphate regulon. Moreover, there was only an overlap of six genes with the gene set responding to severe cytoplasmic phosphate deprivation in the presence of PhoB (Supplementary Data 2) and of one gene (CCNA_01738) with the gene set responding to moderate cytoplasmic phosphate deprivation in the absence of PhoB, suggesting a small extent of dual regulation. These results showed that the analytical pipeline used was able to robustly identify genes that respond to cytoplasmic phosphate deprivation independently of PhoB signaling.

A more detailed analysis of the three cytoplasmic phosphate-regulated gene sets revealed that many genes known to be involved in DNA replication, cell division, stalked-pole morphogenesis and stalk elongation were included either in the core cytoplasmic phosphate regulon or in the set of genes responding to severe phosphate deprivation. In the core set, low

cytoplasmic phosphate levels led to the upregulation of *bacA*, encoding a bactofilin involved in stalk biogenesis[35], and to the down-regulation of the essential cell division gene *fzlA*[36]. Among the gene set regulated upon severe cytoplasmic phosphate starvation (with PhoB present), we observed an upregulation of genes involved in stalk formation, including *staR*[37], *stpX*[38] and *stpA*[39], as well as a downregulation of the genes encoding the DNA replication initiator DnaA[40], the α subunit of DNA polymerase III (DnaE)[41] and the regulator of the essential *dcw* cell wall biosynthesis operon, MraZ[42]. Together, these findings are consistent with the block in DNA replication, the cessation of cell division and the stimulation of stalk elongation induced by phosphate limitation in the Δ*pstS* and, to a lesser extent, the Δ*phoB* mutant.

A gene ontology enrichment analysis based on GO terms revealed the biological functions that were most prevalent among the three gene sets (Supplementary Figs. 8–10 and Supplementary Data 3). The top-scoring functions differed for each group. In the core cytoplasmic phosphate regulon (88 genes), they included processes involved in respiration, amino acid metabolism and the transport of cell envelope precursors (Supplementary Fig. 8). The dominant functions in the gene set regulated under severe cytoplasmic phosphate deprivation (163 genes) were related to detoxification processes, protein and amino acid degradation and the metabolism of storage compounds (Supplementary Fig. 9). By contrast, functions enriched in the gene set regulated only under moderate phosphate starvation (148 genes) were to a large extent related to RNA metabolism and nucleotide biosynthesis. Notably, in the first two gene sets, there is no instance in which most enzymes of a metabolic pathway were coordinately regulated in response to changes in the cytoplasmic phosphate level. Changes in the cytoplasmic phosphate level rather appear to induce global re-adjustments in central cellular pathways related to nucleotide metabolism and cellular stress that enable the cell to cope with the reduced availability of phosphate for energy conservation and biosynthetic processes.

**Table 1 | Characteristics of the indicated strains with respect to the cytoplasmic phosphate level, the activation state of PhoB and the presence of PitA in phosphate-replete conditions**

| Feature | Strain | | | | | |
|---|---|---|---|---|---|---|
| | **WT** | **WT$^{pitA}$** | **Δ$phoB$** | **Δ$phoB^{pitA}$** | **Δ$pstS$** | **Δ$pstS^{pitA}$** |
| Intracellular P$_i$[a] | High | High | Markedly reduced | High | Very low | High |
| PhoB activity[b] | Basal | Basal | OFF | OFF | ON | ON |
| PitA presence | - | + | - | + | - | + |

The strains shown are WT (CB15N), WT$^{pitA}$ (MAB257), Δ$phoB$ (JK2), Δ$phoB^{pitA}$ (MAB258), Δ$pstS$ (JK158) and Δ$pstS^{pitA}$ (MAB259).

[a]Based on the level of *pstSCAB* expression[21,22] and the results of phosphate transport assays performed in this (Fig. 3) and previous[22,25] work.

[b]Based on the results of reporter gene assays (Fig. 6) and transcriptome analyses performed in this (Supplementary Data 1) and previous[21,22] work.

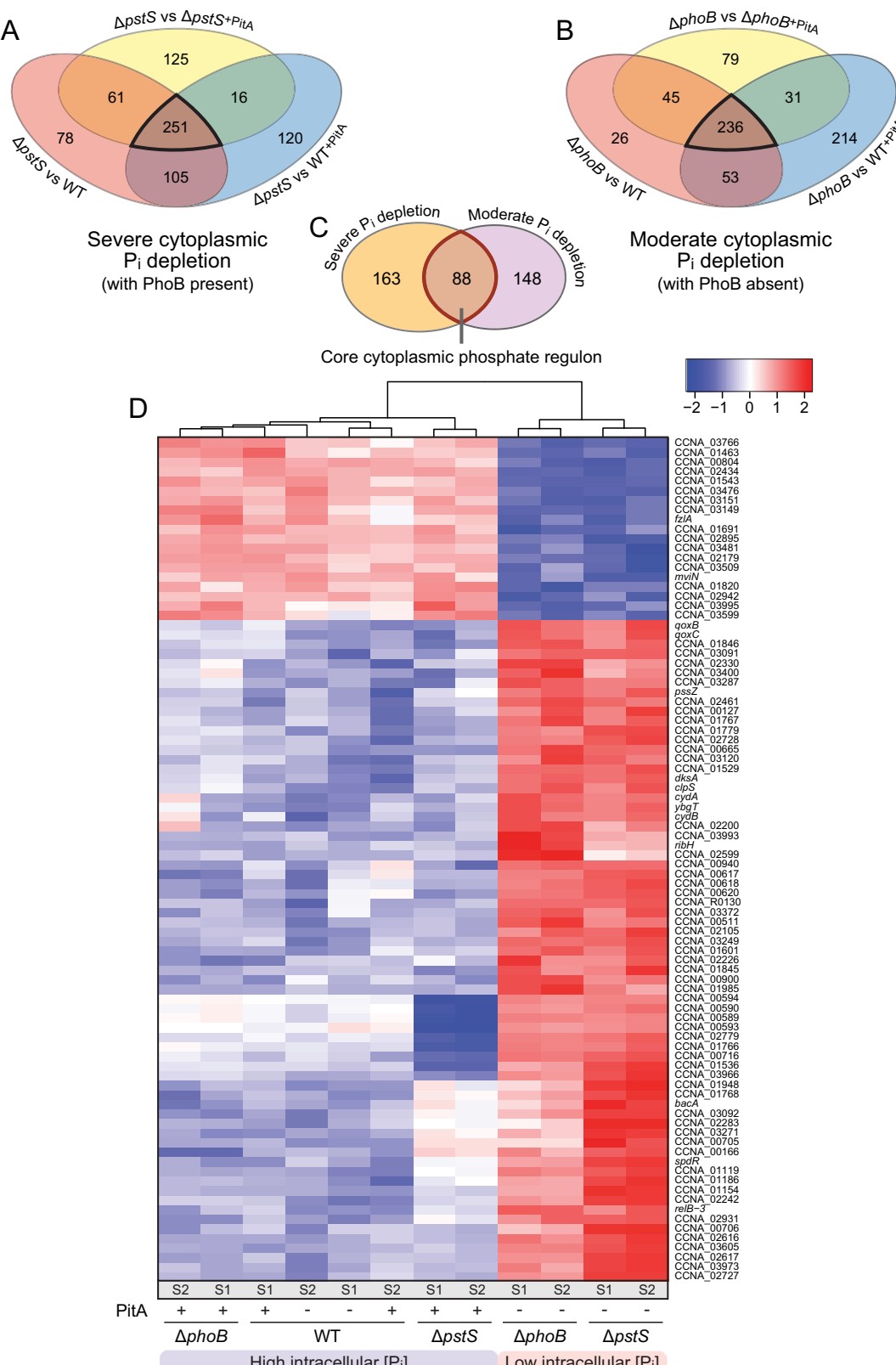

**Fig. 7 | Genes specifically controlled by the cytoplasmic phosphate level. A**, **B** Venn diagrams giving the number of differentially regulated genes that are found in common in pairwise comparisons of cells with severely (ΔpstS; PhoB present) or moderately (ΔphoB) depleted cytoplasmic phosphate pools against cells with high cytoplasmic phosphate levels (WT, WT^pitA, ΔpstS^pitA or WT, WT^pitA, ΔphoB^pitA). Black frames indicate the genes that are significantly regulated in all comparisons shown. The strains analyzed are: ΔpstS (JK158), ΔphoB (JK2), WT (CB15N), WT^pitA (MAB257), ΔpstS^pitA (MAB259), ΔphoB^pitA (MAB258). **C** Core cytoplasmic phosphate regulon, obtained by comparison of the two gene sets defined in (**A**) and (**B**). The red frame indicates the genes that are robustly regulated by changes in the cytoplasmic phosphate concentration, independently of the absence/presence of PhoB. **D** Clustering analysis comparing the expression levels of the 88 genes in the core cytoplasmic phosphate regulon. White color represents the average transcript level of each gene among the tested condition. Red and blue color indicates an increase or decrease, respectively, in the transcript levels compared to the average. Normalized logCPM values were used in each case, leading to a fixed range of values for all genes. S1 and S2 indicate the two replicates analyzed for each strain.

**Fig. 8 | Genes specifically controlled by PhoB.**
**A**, **B** Venn diagrams giving the number of differentially regulated genes that are found in common in pairwise comparisons of cells with highly activated PhoB (Δ*pstS* and Δ*pstS*$^{pitA}$) versus cells with inactive (Δ*phoB*, Δ*phoB*$^{pitA}$) or poorly activated (WT, WT$^{pitA}$) PhoB. Black frames indicate the genes that are significantly regulated in all comparisons shown. The strains analyzed are: Δ*pstS* (JK158), Δ*pstS*$^{pitA}$ (MAB259), Δ*phoB* (JK2), Δ*phoB*$^{pitA}$ (MAB258), WT (CB15N), WT$^{pitA}$ (MAB257). **C** PhoB regulon, obtained by comparison of the two gene sets defined in (**A**) and (**B**). The red frame indicates the genes that are robustly regulated by PhoB, independently of the cytoplasmic phosphate concentration. **D** Clustering analysis comparing the expression levels of the 47 genes in the PhoB regulon. White color represents the average transcript level of each gene among the tested condition. Red and blue color indicates an increase or decrease, respectively, in the transcript levels compared to the average. Normalized logCPM values were used in each case, leading to a fixed range of values for all genes. S1 and S2 indicate the two replicates analyzed for each strain.

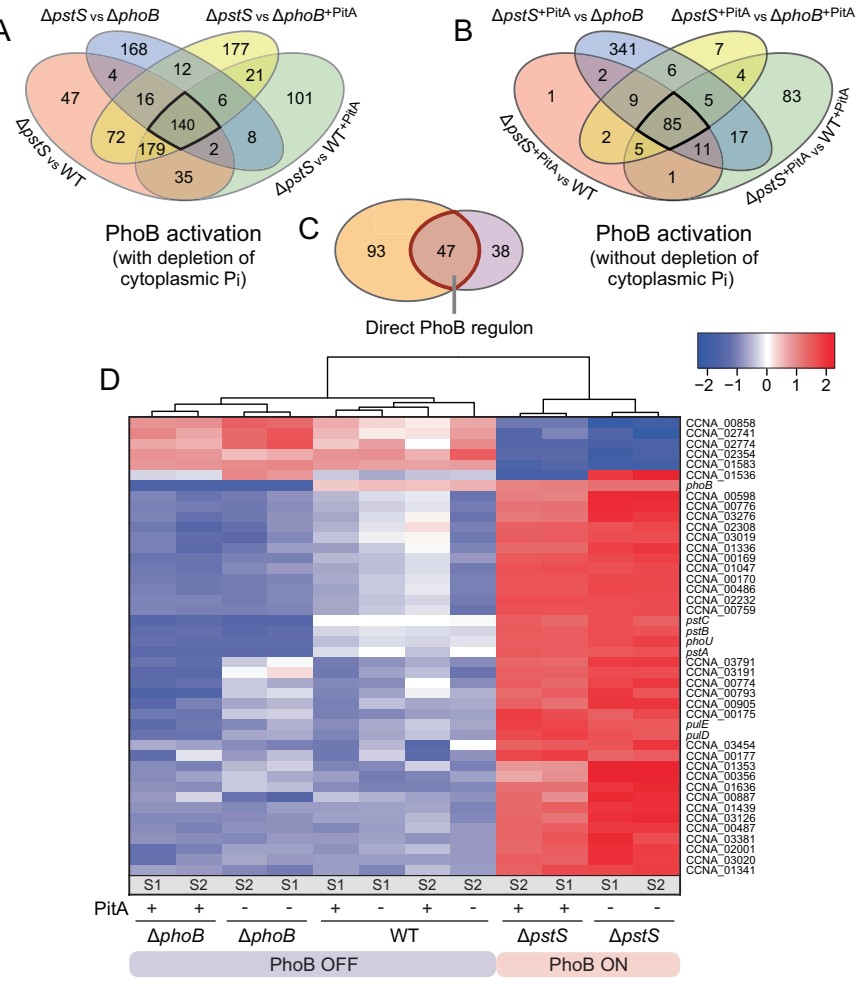

## Identification of the PhoB regulon

Our study demonstrates that *C. crescentus* has a bi-pronged response to phosphate limitation, involving distinct regulatory pathways to sense the extracellular or cytoplasmic phosphate concentration, respectively. Having determined the subset of genes that is controlled by the endogenous phosphate pool, we next aimed to identify the genes whose expression was regulated by the PstSCAB-PhoRB pathway in response to the availability of extracellular phosphate. To this end, we compared the transcriptional profiles of the Δ*pstS* and Δ*pstS*$^{pitA}$ mutants, in which PhoB is fully activated, with those of strains in which PhoB was either completely absent (Δ*phoB* and Δ*phoB*$^{pitA}$) or only activated at basal levels (WT and WT$^{pitA}$) (Supplementary Fig. 5 and Supplementary Data 1). Although all of these comparisons should, in principle, identify members of the PhoB regulon, an analysis of strains expressing *pitA* may be particularly useful for this purpose, because it avoids indirect effects caused by differences in the cytoplasmic phosphate levels. Consistent with this notion, very similar and relatively small sets of differentially regulated genes were obtained by pairwise comparisons of Δ*pstS*$^{pitA}$ cells with WT, WT$^{pitA}$ and Δ*phoB*$^{pitA}$ cells, all of which have high cytoplasmic phosphate levels. The comparisons involving the Δ*pstS* mutant, by contrast, yielded larger differences in the transcriptional profiles, which likely also include differential responses related to the reduced cytoplasmic phosphate level of this strain (Supplementary Data 1).

To define a robust set of PhoB-regulated genes, we again combined multiple comparisons and only considered genes that were differentially regulated in all conditions. The comparison of Δ*pstS* cells with Δ*phoB*, Δ*phoB*$^{pitA}$, WT and WT$^{pitA}$ cells resulted in a total of 140 commonly regulated genes (Fig. 8A). This number was reduced to 85 genes when Δ*pstS*$^{pitA}$ cells were used instead of Δ*pstS* cells as a reference for the comparisons (Fig. 8B). Both gene sets included the *pstC-pstA-pstB-phoU-phoB* operon, which is

known to be expressed under the control of a PhoB-dependent promoter, verifying the validity of the results. A comparison of the two sets identified 47 common genes that could be assigned with high confidence to the (direct or indirect) PhoB regulon and were not subject to regulation by the cytoplasmic phosphate concentration (Fig. 8C). A clustering analysis based on these genes revealed two distinct groups of transcriptomes, one from cells in which PhoB is fully active and another one from cells in which PhoB is absent or only active at basal levels (Fig. 8D). A comparison with a previously established list of direct PhoB targets that was obtained by a combination of transcriptome and ChIP-seq analysis[22] identified 24 common genes, which can thus be regarded with high confidence as members of the direct regulon of PhoB (Supplementary Data 2). The remaining genes may be regulated by PhoB in an indirect fashion. Notably, previously reported direct PhoB targets[22] that are not included in the present list of PhoB-regulated genes typically show only weak responses to phosphate limitation, so that they may not have fulfilled the stringent criteria set to defined the PhoB regulon in this study.

A gene ontology analysis of the members of the PhoB regulon (47 genes) revealed a high prevalence of functions related to protein secretion, ion transport and the degradation of phosphate-containing compounds (Supplementary Fig. 11 and Supplementary Data 3). These categories included, for instance, the type II secretion system of *C. crescentus*, which has previously been shown to be upregulated during phosphate starvation, promoting indirectly the release of a lipoprotein (ElpS) that stimulates extracellular alkaline phosphatase activity[43]. They also included the PstSCAB transport system, various TonB-dependent receptors, likely mediating the uptake of alternative phosphate donors, as well as enzymes predicted to be involved in the degradation of phosphate-containing molecules, such as exonucleases, a nucleotide pyrophosphatase, a phytase, various phosphatases and a phosphodiesterase. *C. crescentus* has recently

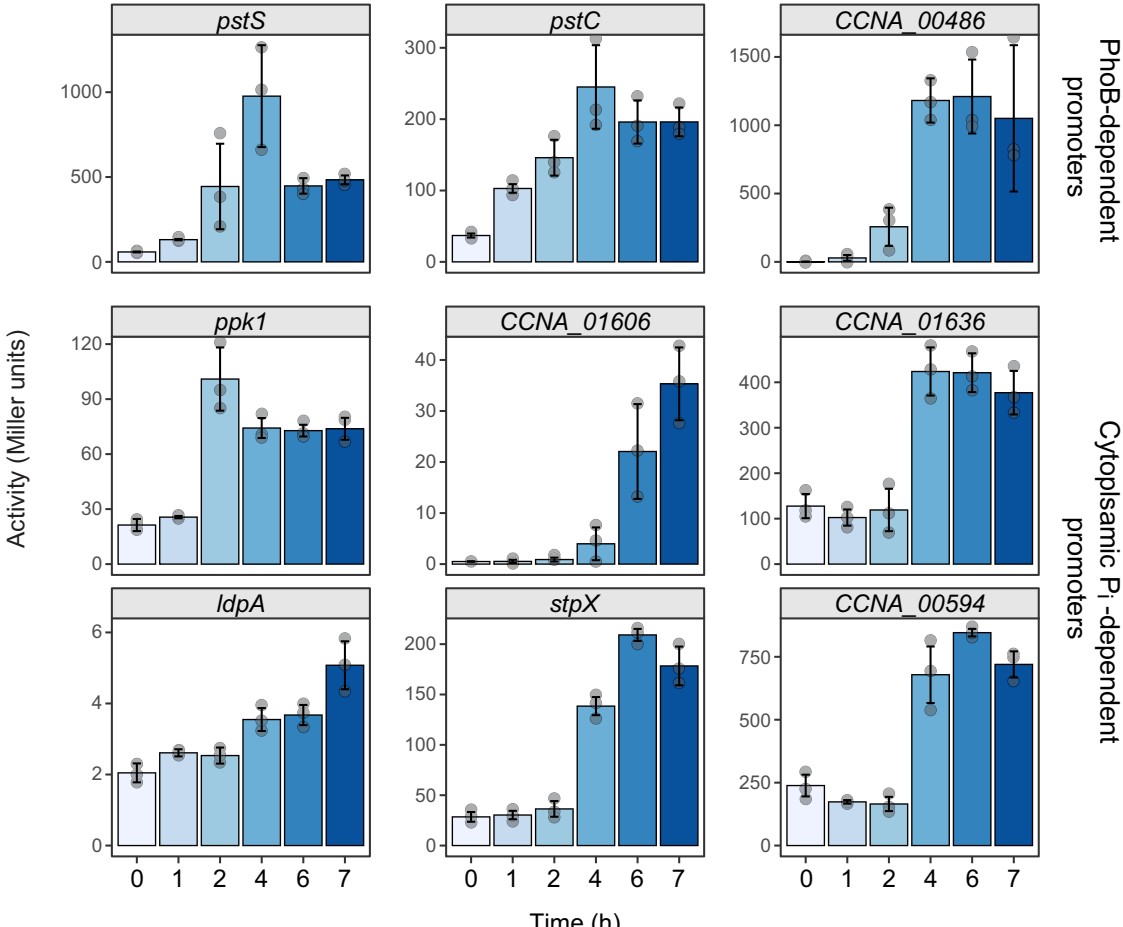

**Fig. 9 | Induction kinetics of phosphate-regulated genes upon phosphate deprivation.** Wild-type (CB15N) cells harboring reporter plasmids that carry fusions of the *pstS* (pMAB34), *pstC* (pMAB218), *CCNA_00486* (pMAB105), *ppk1* (pMAB106), *CCNA_01606* (pJR16), *CCNA_01636* (pMAB226), *ldpA* (pMAB219), *stpX* (pMAB222) and *CCNA_00594* (pMAB223) promoter regions to the *lacZ* gene were grown in PYE medium and then diluted (1:20) into M2G$^{-P}$ medium. Samples were taken at the respective time points and subjected to β-galactosidase activity assays. Data represent the mean (± SD) of three independent measurements.

been shown to dramatically change its membrane composition during phosphate starvation by degrading its phospholipids and replacing them with glycolipids and glycosphingolipids[44]. Consistent with this observation, the PhoB regulon also included genes for phospholipid-degrading enzymes (glycerolphosphodiester phosphodiesterases, phospholipase C) as well as for enzymes involved in glycosphingolipid biosynthesis (ceramide synthase, ceramide glycosyltransferase). Overall, PhoB-regulated genes thus appear to be mainly involved in the mobilization of phosphate from extracellular and endogenous phosphate-containing compounds as well as the uptake of free inorganic phosphate from the environment.

### Cytoplasmic phosphate-controlled genes show a delayed response to phosphate deprivation

Our transcriptome analyses identified two functionally distinct gene sets that are controlled by the cytoplasmic phosphate level or the availability of extracellular phosphate, as sensed by the PstSCAB-PhoRB pathway, respectively. Notably, only seven genes (CCNA_00598, CCNA_00356, CCNA_01353, CCNA_00776, CCNA_03126, CCNA_01636, CCNA_02001) are found both in the PhoB regulon and in the list of genes regulated under severe cytoplasmic phosphate deprivation (163 genes; see Fig. 7C), and only one gene (CCNA_01536) of the PhoB regulon is also included in the core cytoplasmic phosphate regulon (88 genes; see Fig. 8C), suggesting a relatively low degree of dual regulation. This small overlap underscores the strict regulatory and functional separation of PhoB- and cytoplasmic phosphate-regulated genes in *C. crescentus*.

Given that PhoB upregulates various genes involved in the mobilization of intracellular phosphate sources, it was conceivable that its activation upon phosphate starvation allows cells to maintain cytoplasmic phosphate homeostasis for a prolonged period of time until the endogenous supplies are depleted. To test this hypothesis, we monitored the promoter activities of various genes controlled by either PhoB or the cytoplasmic phosphate level after the dilution of cells into phosphate-free (M2G$^{-P}$) medium. The results showed that all PhoB-dependent promoters showed an immediate response to the drop in the extracellular phosphate concentration. The promoters controlled by the cytoplasmic phosphate level, by contrast, were upregulated asynchronously at later time points, likely responding to different degrees of cytoplasmic phosphate deprivation (Fig. 9). *C. crescentus* thus adapts to phosphate deprivation in a biphasic process, driven by the expression of two distinct sets of genes that adjust cellular physiology to the successive depletion of environmental and cytoplasmic phosphate, respectively.

### Discussion

Phosphate availability is one of the key parameters that determines bacterial growth and survival in soil and oligotrophic aquatic habitats[45,46]. So far, studies investigating the response of bacteria to phosphate limitation have mainly focused on the highly conserved PstSCAB-PhoRB pathway, which is thought to sense the concentration of extracellular phosphate and induce the production of proteins that promote the mobilization and assimilation of phosphate from various organic and inorganic sources under phosphate limitation[1].

In this study, we show that the fresh-water bacterium *C. crescentus* specifically senses and responds to changes in the cytoplasmic phosphate level in a manner independent of the PstSCAB-PhoRB pathway. Our work was enabled by the heterologous production of the phosphate transporter PitA from *E. coli* in *C. crescentus* cells, which made it possible to uncouple the accumulation of cytoplasmic phosphate from the transport activity of the PstSCAB system. Using this approach, we found that the PhoR-PhoB signaling pathway of *C. crescentus* is exclusively controlled by extracellular phosphate sensing through the PstSCAB system and unaffected by the cytoplasmic phosphate level, suggesting the lack of a negative feedback mechanism to regulate PstSCAB activity or PhoB activation. This finding is in stark contrast to results obtained in the enteric bacteria *E. coli* and *Salmonella enterica*, which suggest that in these species the cytoplasmic phosphate level may have a decisive role in the regulation of PhoB activity, with PstSCAB, PhoR and PhoU constituting an intracellular phosphate-sensing system that functions independently of the transport activity of the PstSCAB system[30,31,47]. Thus, different species may have evolved distinct phosphate-sensing strategies to optimize fitness in the environmental niches they inhabit. The genetic approach presented in this study may provide a straightforward means to investigate the underlying regulatory mechanisms.

Our transcriptome analyses show that there is a clear separation of the gene sets regulated by PhoB and the cytoplasmic phosphate level, respectively, with only little overlap. PhoB controls the production of various TonB receptors, which may facilitate the transport of phosphate-containing compounds across the outer membrane and thus promote their uptake into the cytoplasm as a source of phosphate for metabolic processes. Moreover, it regulates systems to mobilize phosphate from phosphate-containing extracellular or endogenous molecules such as nucleic acids and phospholipids. Its activation thus appears to induce pathways that allow the cell to utilize alternative phosphate sources and exploit its endogenous phosphate reservoirs. The cytoplasmic phosphate level, by contrast, controls the production of various metabolic enzymes as well as factors involved in DNA replication, cell division and stalked-pole development, thereby adjusting cellular metabolism, the cell cycle and cell morphogenesis to the reduced availability of phosphate in the cytoplasm. These responses may be further graded according to the degree of cytoplasmic phosphate deprivation, with different pathways affected at different phosphate concentrations. The dichotomy between extracellular and cytoplasmic phosphate sensing enables cells to differentiate between the supply of free inorganic phosphate in the environment, as sensed by the PstSCAB-PhoRB pathway, and the total amount of phosphate that is available for cellular metabolism. When free inorganic phosphate is depleted in the environment, cells will activate PhoB to promote the utilization of alternative phosphate sources and thus maintain phosphate homeostasis in the cytoplasm over a prolonged period of time. Only when these alternative sources are consumed and the cytoplasmic level drops below a critical threshold, global readjustments are induced to prepare the cells for growth and persistence under global phosphate deprivation.

The mechanism responsible for cytoplasmic phosphate sensing is still unclear. A potential sensor of free endogenous phosphate could be PhoU, which was shown to be essential for viability and critical for proper phosphate metabolism in *C. crescentus*[22]. However, the depletion of PhoU did not interfere with the restoration of normal stalk length or growth in a Δ*pstS* mutant when its cytoplasmic phosphate pool was replenished by the production of PitA (Supplementary Fig. 12), suggesting that it may not have a major role in the sensing mechanism. Alternatively, there may be multiple independent regulatory systems that monitor either the availability of free inorganic phosphate in the cytoplasm or, as an indirect readout, the levels of metabolites that are formed in phosphate-dependent enzymatic reactions. The coordination of the responses to extracellular and cytoplasmic phosphate starvation may be facilitated by the dual regulation of target genes—a situation that was observed for at least eight genes in our study.

Interestingly, our results demonstrate that the characteristic cell and stalk elongation phenotype of *C. crescentus* in phosphate-limited media is not triggered directly by PhoB signaling but specifically induced by the depletion of the cytoplasmic phosphate pool. This finding is consistent with previous results showing that *C. crescentus* cells producing a mutant variant of the phosphomannose isomerase ManA failed to elongate their stalk upon phosphate starvation, although PhoB was still activated under these conditions[48]. Thus, phosphosugar metabolism and the availability of cytoplasmic phosphate to form these sugars appear to play a critical role in stalk elongation, independently of PhoB activation. The stalk elongation defect of the Δ*phoB* mutant in phosphate-limited media may be an indirect consequence of its reduced growth and fitness during phosphate starvation. This phenotype may be related to a defect in the utilization of alternative phosphate sources, caused by the disruption of PhoB signaling. However, the molecular pathways linking the cytoplasmic phosphate pool to stalk biosynthesis still remain to be determined.

Collectively, our results provide a new view of how *C. crescentus* senses and reacts to the availability of phosphate. They demonstrate that the regulatory processes involved in the adaptation of bacterial cells to phosphate limitation are more complex than previously thought and involve sensing mechanisms that go beyond the classical PhoRB signaling pathway. It will be interesting to use the heterologous expression of PitA to study the contributions of extracellular and cytoplasmic phosphate sensing to the overall phosphate starvation response in other bacterial species.

## Methods
### Media and growth conditions
*Caulobacter* NA1000 (Evinger & Agabian[49]) and its derivatives were grown at 28 °C in peptone-yeast-extract (PYE) medium[50] (0.2% Bacto Peptone, 0.1% Bacto Yeast Extract, 1 mM $MgSO_4$, 0.5 mM $CaCl_2$; containing ~0.2 mM inorganic phosphate derived from peptone and yeast extract[51]), M2G medium (6.1 mM $Na_2HPO_4$, 3.9 mM $KH_2PO_4$, 9.4 mM $NH_4Cl$, 0.2% [w/v] glucose, 0.5 mM $MgCl_2$, 0.5 mM $CaCl_2$, 0.01 mM $FeSO_4$/0.01 mM EDTA) or $M2G^{-P}$ medium[35] (M2G medium containing 20 mM Tris/HCl pH 7.0 instead of phosphate salts). Media were supplemented with antibiotics at the following concentrations when appropriate (µg ml$^{-1}$; liquid/ solid medium): kanamycin (5/25), chloramphenicol (1/1), oxytetracycline (1/2). To induce phosphate starvation, stationary cultures were diluted 1:20 in $M2G^{-P}$ medium (resulting in an $OD_{600}$ of ~0.06) and incubated at 28 °C for the indicated times. The expression of the *pitA* gene from the *xylX* promoter (P*xyl*) was induced by supplementation of the media with 0.3% D-xylose. The conditional *phoU* mutant was kept in media supplemented with 0.5 mM sodium vanillate to achieve the expression of *phoU*. For growth analyses, cells were cultivated to the exponential or early stationary phase in PYE medium and then washed three times with phosphate-free medium or diluted (1:20) into PYE, M2G or $M2G^{-P}$ medium, as indicated. The suspensions were transferred to 24-well polystyrene microtiter plates (Becton Dickinson Labware), incubated at 32 °C with double-orbital shaking in an Epoch 2 microplate reader (BioTek, Germany), and analyzed photometrically ($OD_{600}$) at 15 min intervals.

*E. coli* strain TOP10 (Invitrogen) and its derivatives were cultivated at 37 °C in LB broth (Karl Roth, Germany). Antibiotics were added at the following concentrations (µg ml$^{-1}$; liquid/solid medium): kanamycin (30/ 50), chloramphenicol (20/30), oxytetracycline (12/12).

### Plasmid and strain construction
The bacterial strains, plasmids, and oligonucleotides used in this study are listed in Tables S1–S4. *E. coli* TOP10 (Invitrogen) was used as host for cloning purposes and as a source of the *pitA* gene. *E. coli* was transformed by chemical transformation and *C. crescentus* was transformed by electroporation. Non-replicating plasmids were integrated at the chromosomal *xylX* (P*xyl*) locus of *C. crescentus* by single-homologous recombination[52]. Gene replacement was achieved by double-homologous recombination using the counter-selectable *sacB* marker[53], and proper chromosomal integration or gene replacement was verified by colony PCR.

## Light microscopy

For light microscopic analysis, cells were transferred onto pads made of 1% agarose. Images were taken with an Axio Observer.Z1 (Zeiss) microscope equipped with a Plan Apochromat 100×/1.45 Oil DIC and a Plan Apochromat 100×/1.4 Oil Ph3 phase contrast objective, and a pco.edge sCMOS camera (PCO). Images were recorded with VisiView 3.3.0.6 (Visitron Systems, Germany) and processed with Fiji[54] and Illustrator CS6 (Adobe Systems, USA). Stalk lengths and cell dimensions were measured with the image analysis software BacStalk[55].

## Phosphate uptake assay

To determine the kinetic parameters of the uptake of inorganic phosphate, cells of different *C. crescentus* strains were grown in PYE medium supplemented with 0.3% xylose to ensure the expression of genes placed under the control of the P*xyl* promoter, washed three times with M2G$^{-P}$ medium and then inoculated into fresh M2G$^{-P}$ medium supplemented with xylose. After incubation at 30 °C for 6 h, the cells were washed again once to remove residual phosphate. Subsequently, the optical densities of the samples were measured, and uptake assays were started by mixing aliquots (250 µl) of the cell suspensions with various concentrations of radiolabeled phosphate. The phosphate concentrations used in the uptake assay varied between 0.1 µM and 20 µM. All reactions contained a constant concentration of 0.5 nM [$^{33}$P] phosphate as a tracer, resulting in specific activities between 0.7 and 140 Ci/ mmol. [$^{33}$P]phosphate (40–158 Ci/mg) was purchased from Perkin Elmer (Rodgau-Jügesheim, Germany). The assay mixtures were incubated at 37 °C. To follow phosphate uptake over time, cell samples (50 µl) were taken at 15-s intervals and filtered through cellulose filters with 2.5 cm diameter and pore sizes of 0.45 µm (ME25, GE Healthcare, Freiburg, Germany) that were pre-wetted with a 100 mM potassium phosphate solution (pH 7.3). The filters were washed with 20 ml H$_2$O and the radioactivity retained on the filters was determined by scintillation counting (TriCarb B2810, PerkinElmer, Rodgau-Jügesheim, Germany). The amounts of intracellular phosphate measured were normalized to the optical densities of the samples and expressed as nmol*OD$_{600}$ units$^{-1}$, with an OD$_{600}$ unit corresponding to the amount of cell material that is contained in 1 ml of a culture with an OD$_{600}$ value of 1. The data obtained were fitted to the Michaelis-Menten equation using Prism 6 software (GraphPad, San Diego, CA).

## β-Galactosidase assays

β-Galactosidase assays were conducted at 28 °C as described previously[56]. Briefly, 1 ml of cells of known OD$_{600}$, grown in PYE medium supplemented with 0.3% xylose, were lysed by the addition of 100 µl chloroform and 50 µl 0.1% SDS and subsequent vigorous shaking. 500 µl of Z buffer (60 mM Na$_2$HPO$_4$, 40 mM NaH$_2$PO$_4$, 10 mM KCl, 1 mM MgSO$_4$, 50 mM β-mercaptoethanol) were added to 500 µl of permeabilized cells to reach a total volume of 1 ml. Cells grown in M2G$^{-P}$ medium were analyzed using a modified, phosphate-free Z buffer (100 mM Tris-HCl pH 7.2, 10 mM KCl, 1 mM MgSO$_4$, 50 mM β-mercaptoethanol). The suspension was then mixed with 200 µl of ONPG solution (4 mg ml$^{-1}$ *o*-nitrophenyl-β-D-galactopyranoside in Z buffer) to start the reaction. Once a yellow color had developed, the reaction was stopped with 500 µl of 1 M Na$_2$CO$_3$. The OD$_{420}$ and OD$_{550}$ of the supernatant were measured and β-galactosidase activities (in Miller Units; MU) were calculated using the equation MU = 100*(OD$_{420}$-1.75*OD$_{550}$)/(OD$_{600}$*time [in min]* volume of culture [in ml]). The reported data represent the averages of three independent experiments.

## RNA-seq experiment and transcriptome data analysis

Each of the six strains analyzed was grown (in two biological replicates) in PYE medium supplemented with 0.3% xylose and no antibiotics (to avoid any adverse effects on the expression profiles) to the mid-exponential growth phase (OD$_{600}$ = 0.3–0.5). Due to the slow growth of Δ*phoB* and Δ*pstS* cells, samples for these strains were collected at later time points than for the other strains. Cells were harvested by centrifugation, snap-frozen in liquid nitrogen and stored at −80 °C. RNA extraction, cDNA library preparation, and sequencing were performed by Vertis Biotechnologie AG (Freising, Germany), which provided both the raw reads and normalized data (available at the GEO database under accession number GSE244776) (Supplementary Data 1).

All subsequent analyses, including sample normalization and comparisons, were conducted with the EdgeR package (version 3.43.4) of the Bioconductor suite (release 3.17)[57,58]. EdgeR was supplied with the raw counts as specified in the instructions. After the replicates of each strain had been clustered and normalized, pairwise comparisons of the strains were performed initially with a quasi-likelihood F-test (glmQLFTest), yielding the false discovery rate (FDR) values, and then further refined with glmTreat using a fold change threshold (LogFC > 1.5). MDS plots were generated using the respective function included in EdgeR. All other plots were generated in R version 4.3.1[59] using the ggplot2 package[60]. Heatmaps were generated with heatmap.2 in R as described previously (Chen et al.[61]), by first employing logCPM values derived from EdgeR and subsequently normalizing them with the scale function in R. Gene ontology enrichment analysis was performed with the topGO package v2.52.0[62], available on the Bioconductor website (www.bioconductor.org). The software was provided with the locus tags of preselected gene groups, already considering the thresholds for the FDR and LogFC values. The subsequent statistical analysis was based on a classical Fisher test.

## Statistics and reproducibility

All experiments were performed at least twice independently with similar results, unless indicated otherwise. No data were excluded from the analyses. Calculations and statistical analyses were performed in R 4.3.1[59]. The statistical significance of differences between datasets was determined using unpaired two-sided Welch's *t* tests. To quantify imaging data, multiple images were analyzed per condition. The analyses included all cells in the images or, in the case of high cell densities, all cells in a square portion of the images. The selection of the images and fields of cells analyzed was performed randomly.

## Availability of biological material

The plasmids and strains used in this study are available from the corresponding author upon request.

## Reporting summary

Further information on research design is available in the Nature Portfolio Reporting Summary linked to this article.

## Data availability

The RNA-seq data have been deposited at the the Gene Expression Omnibus (GEO) database with the accession number (GSE244776). All other data generated in this study are included in the manuscript or the supplemental information. Source data and uncropped images are provided in Supplementary Data 4.

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

## Acknowledgements

We thank Julia Rosum for excellent technical assistance. This work was funded by the University of Marburg (core funding to M.T. and E.B.) and the Max Planck Society (Max Planck Fellowship to M.T.).

## Author contributions
M.B. and J.K. constructed plasmids and strains. M.B. performed the growth analyses, reporter gene assays and RNA-seq analyses. T.H. performed the phosphate uptake assays. M.B. and T.H. analyzed the data. E.B. and M.T. supervised the study and acquired funding. M.B. and M.T. conceived the study and wrote the manuscript, with input from all other authors.

## Funding

## Competing interests
The authors declare no competing interests.
