## [Peer Review File · Communications Biology]

Reviewers' comments:

Reviewer #1/#5 (Remarks to the Author):

In this manuscript, Billini et al. investigate the response to phosphate starvation in *C. crescentus*. By using an elegant approach to uncouple PhoB signaling from phosphate uptake, they identify genes that are dependent on intracellular phosphate levels and independent of PhoB. The paper does not address the mechanistic basis of how *C. crescentus* might be sensing cytoplasmic phosphate levels. However, even in the absence of these more mechanistic data, the paper is a valuable contribution to the field. The conclusions drawn in this study are generally well supported by the data and the paper is well-written.

Specific comments:

- In line 75-76 the authors reference the fact that PhoU has been suggested to sense intracellular phosphate level, however there is no investigation of whether PhoU is the unidentified sensor. It would strengthen this manuscript to further investigate if PhoU is important for this pathway, particularly as there is already a depletion strain available (Lubin et al. 2015, J. Bact.).
- Lines 53-55: The authors have stated that phosphate starved cells are dormant, however cell size continues to increase under this condition indicating that cells are still metabolically active. The authors should reword this statement and avoid the word dormant in this context.
- Line 57: The statement that *C. crescentus* does not possess any alternative phosphate transport systems is quite strong. This should be reworded to either "does not seem" or "has no other characterised phosphate transporters". It should also be clearer that there are other transporters for phosphorylated compounds.
- Figure S2: The authors have chosen to grow their cells in PYE to stationary phase before diluting 1:20 into either PYE or M2G-P. As a control for this experiment, dilution in nutrient replete M2G should also be included. This is to account for the differences that may happen when diluting from complex media to a defined glucose-based minimal medium. Additionally, consideration should be made for the fact that the P starvation experiments were done with pre-cultures grown to stationary phase. Different strains may reach stationary phase with different timing and thus have been cultured in stationary phase for different durations of time. This may affect how they respond to P starvation following dilution.
- Figure S3B: The data in this figure is simply the CFU/ml count at the time point taken, however it cannot be used to estimate the relative fitness of each of the strains. The *phoB* mutant likely has a lower CFU/ml count because it was taken from a culture with a much lower OD.
- Figure 2: Normalisation should be to OD and volume rather than to assumed protein amounts.
- Figure 3 and Figure S1 the authors should consider to include quantification of the cell lengths in addition to the stalk lengths as the increase in cell size is also indicative of phosphate starvation.
- Figure 4, Figure S2, Figure S3A: The scale for the growth curves should be in log scale.
- Figure 4: It would be helpful to have the PitA minus and plus plotted on the same axis to more clearly compare the conditions.
- Figure 7: In panel A the *d_pstS*+PitA vs WT appears to be mislabelled, should it not be *d_pstS* vs WT+PitA? This figure is very difficult to make sense of and perhaps panels A and B could be more clearly labelled.

- Table 1: What is the source of the intracellular phosphate levels? Is this experimentally validated, an assumption or from literature? The authors should make this clearer and include references if needed.

Reviewer #2 (Remarks to the Author):

In this report by Billini et al, the authors investigate distinct pathways regulated by intracellular and extracellular phosphate availability in *C. crescentus*. The canonical phosphate uptake system in *C. crescentus*, consisting of PstSCAB, regulates the activity of the PhoB two component system to induce expression of phosphate-regulated genes. A clear phenotype of phosphate limitation, due to low exogenous phosphate levels or deletion of the *pstS* phosphate, is the extension of the polar stalk. While the Pst system monitors environmental phosphate concentrations, other bacteria appear to have systems to simultaneously assess cytoplasmic phosphate levels. Unlike other organisms which have multiple phosphate uptake systems, *Caulobacter* appears to exclusively use the Pst transporter. This makes it a good model system to investigate the distinct roles of extracellular and cytoplasmic phosphate in regulating gene transcription. The authors exogenously expressed the PitA transporter from *E. coli* to enable phosphate uptake in a *pstS* mutant. This strain would have high intracellular phosphate, but would be unable to activate PhoB. Using a series of pair-wise RNAseq experiments, the authors identified a set of genes regulated by cytoplasmic phosphate independent of PhoB. They also identified genes specifically under the control of PhoB. Interestingly, these were non-overlapping sets of genes. Most surprisingly, they found that stalk elongation, the most characteristic phenotype of phosphate limitation, is not regulated by PhoB.

Overall, this paper uses a very nice genetic system to tease apart the various mechanisms underlying the *Caulobacter* response to changes in phosphate availability. The conclusions are consistent with the data presented and I have no major concerns.

I do have a couple of suggestions that may help strengthen the paper further.

1. Since *Caulobacter* does not have a second low-affinity phosphate transporter, it would seem that PstS is always responsible for phosphate import. Yet only when levels drop dramatically does it induce PhoB activation. If so, would the "cytoplasmic" phosphate sensor be triggered at concentrations higher than PhoB. What would happen if cells were grown in M2G (or HIGG) with varying concentrations of phosphate. Your data predict that there should be an order to which genes are turned on first as phosphate levels drop. That could be tested by qRT-PCR using candidate genes from the RNAseq experiments.

2. Given the GO terms for genes regulated by the core vs severe phosphate limitation, you might use metabolomics to show that the predicted metabolic pathways are differentially regulated under these different phosphate regimes.

Reviewer #3 (Remarks to the Author):

Billini et al. provide a detailed characterization of the physiological and transcriptional response of *C. crescentus* to changing phosphate levels. This is important, given the crucial role that phosphorus and phosphate availability plays in various essential cellular processes that support growth and proliferation. Using targeted deletion mutants and the heterologous expression of an *E. coli* phosphate transporter, they are able to disentangle contributions from environmental and cytoplasmic phosphate levels. With RNA sequencing experiments and downstream analyses comparing different strains, they show that a specific response is mounted to altered cytoplasmic phosphate levels, independent of the

PhoR/PhoB two-component system and the PstSCAB transport system. The study is the first to identify the specific sensing and responding to alterations in cytoplasmic phosphate level in *C. crescentus*. In general, the study is well written and the results, figures, and discussion are clearly presented and interesting. The findings likely also have broader relevance, given that all bacteria face the challenge of modulating their growth and proliferation to current nutrient availabilities (which includes phosphate). Below, I have listed some (minor) aspects that I would like to see resolved.

Main comments

-The first section of the manuscript that focuses on the characterization (stalk elongation + growth) of the *phoB* deletion mutant appears a bit out of place and not in line with the second section. I understand that the goal is to set the stage somewhat for later experiments but wonder if these results cannot simply be incorporated there.

-The morphological characterizations currently focus solely on stalk lengths, while other factors (e.g., cell width and length) also appear to be affected. In addition, based on the transcriptional response to altered cytoplasmic phosphate levels, can the authors speculate on any of the molecular mechanisms that underlie cell morphological alterations in conditions where phosphate is limiting?

-Despite that there is limited overlap between the identified PhoB regulon and the core cytoplasmic phosphate regulon, the authors still mention an interplay between extracellular and intracellular phosphate sensing? To me, it is currently unclear how this interplay would be established (and what would be the importance of this).

-Line 328-331: how do the authors explain the differences in the *phoB* regulon that they identified and the one that was previously characterized. Do differences in experimental approach play a role? If so, it would be nice if the authors could elaborate on this to make this more clear.

-Given that PhoB typically functions as a transcriptional activator (or at least, that is how its presented in Figure 1), how do the authors explain the subset of genes that is downregulated when PhoB is ON (Figure 7)?

Minor comments

-Line 41: "the" appears to be missing in between "In this model," and "phosphate-loaded state..."

Reviewer #4 (Remarks to the Author):

Overall, the manuscript is well-written and addresses an interesting (and previously outstanding) question in the field (mostly *Caulobacter* folks). That is, how does *Caulobacter* regulate phosphate uptake (extracellularly and intracellularly) and how does this relate to cell shape and morphology? In terms of methodology, it appears that the authors have performed detailed and robust analyses, which lead to strong, novel findings. I do worry about the utility of these findings across a broader scope. But I think that the work should be accepted for publication. The authors should, however, revise their Results section to either (a) omit much of the discussion points or (b) include the important parts from the Discussion section into the Results section. In its current form, the Results section reads more like a Results + Discussion section. Similarly, the Introduction section is a bit lengthy and could benefit from some concision.

We thank the five reviewers for the positive evaluation of our manuscript and we appreciate their constructive criticism and helpful comments, which helped to further improve our study. Please see below for our responses to the issues raised.

Reviewers' comments:

Reviewer #1/#5

In this manuscript, Billini et al. investigate the response to phosphate starvation in *C. crescentus*. By using an elegant approach to uncouple PhoB signaling from phosphate uptake, they identify genes that are dependent on intracellular phosphate levels and independent of PhoB. The paper does not address the mechanistic basis of how *C. crescentus* might be sensing cytoplasmic phosphate levels. However, even in the absence of these more mechanistic data, the paper is a valuable contribution to the field. The conclusions drawn in this study are generally well supported by the data and the paper is well-written.

Specific comments:

- In line 75-76 the authors reference the fact that PhoU has been suggested to sense intracellular phosphate level, however there is no investigation of whether PhoU is the unidentified sensor. It would strengthen this manuscript to further investigate if PhoU is important for this pathway, particularly as there is already a depletion strain available (Lubin et al. 2015, J. Bact.).

We have now investigated whether PhoU could have a role in sensing the cytoplasmic phosphate level (new Figure S12). Our results show that in the absence of PstS and in the presence of PitA as the sole phosphate transport system, cells display similar stalk lengths and growth behaviors with or without PhoU, indicating that PhoU does not act as a global regulator of the cytoplasmic phosphate starvation response. Notably, the expression of *phoU* is positively regulated by PhoB (see Figure 1). However, we expect that the cytoplasmic phosphate sensor is not subject to PhoB regulation, since our work shows that the response to cytoplasmic phosphate deprivation occurs independently of the activation state of the PhoB. Overall, the precise role of PhoU in *C. crescentus* remains incompletely understood, but a detailed analysis of its function is beyond the scope of the present study.

- Lines 53-55: The authors have stated that phosphate starved cells are dormant, however cell size continues to increase under this condition indicating that cells are still metabolically active. The authors should reword this statement and avoid the word dormant in this context.

Thank you for pointing out the wrong wording. We now state that the cells enter a “non-proliferative state”.

- Line 57: The statement that *C. crescentus* does not possess any alternative phosphate transport systems is quite strong. This should be reworded to either “does not seem” or “has no other characterised phosphate transporters”. It should also be clearer that there are other transporters for phosphorylated compounds.

The text has been changed as follows: “... this species does not appear to possess any other high-affinity phosphate uptake system, such as homologs of the *E. coli* PitA or PitB proteins, although it contains transporters for alternative phosphorus sources such as phosphonates”.

- Figure S2: The authors have chosen to grow their cells in PYE to stationary phase before diluting 1:20

into either PYE or M2G-P. As a control for this experiment, dilution in nutrient replete M2G should also be included. This is to account for the differences that may happen when diluting from complex media to a defined glucose-based minimal medium. Additionally, consideration should be made for the fact that the P starvation experiments were done with pre-cultures grown to stationary phase. Different strains may reach stationary phase with different timing and thus have been cultured in stationary phase for different durations of time. This may affect how they respond to P starvation following dilution.

To account for differences that may arise after diluting from complex to defined minimal medium, we have now also analyzed the growth of wild-type, $\Delta phoB$ and $\Delta pstS$ cells after dilution from PYE medium into M2G medium (containing 10 mM phosphate). After transfer to M2G medium, the three strains showed behaviors similar to that in PYE medium, with $\Delta phoB$ cells growing slightly slower and $\Delta pstS$ cells growing markedly slower than wild-type cells (new Figure 2B). These results demonstrate that the distinct behavior of wild-type, $\Delta phoB$ and $\Delta pstS$ cells in M2G^{-P} medium is largely based on their different abilities to cope with phosphate starvation and not on differences in their ability to adapt to minimal medium in general.

To clarify the effect exerted by the growth state of the cells prior to their transfer to M2G^{-P} medium, we included additional growth experiments using exponentially growing wild-type, $\Delta phoB$ and $\Delta pstS$ cells as an inoculum. The growth profiles obtained were qualitatively similar to those obtained for cultures inoculated with stationary cells, with $\Delta phoB$ cells being unable to grow efficiently and $\Delta pstS$ resuming wild-type-like growth after an extensive lag phase (new Figure 2D).

- Figure S3B: The data in this figure is simply the CFU/ml count at the time point taken, however it cannot be used to estimate the relative fitness of each of the strains. The *phoB* mutant likely has a lower CFU/ml count because it was taken from a culture with a much lower OD.

To address this point, we transferred equal amounts of wild-type and $\Delta phoB$ cells (diluted or washed) into in M2G^{-P}, incubated them continuously for 72 h and then determined the relative numbers of colony-forming units. In this modified experimental setup, $\Delta phoB$ cells again showed significantly lower CFU/ml counts than wild-type cells (new Supplementary Figure 2B). This finding shows that PhoB is required to grow efficiently in phosphate-limiting conditions, supporting the conclusion that the $\Delta phoB$ mutant has a lower fitness (as defined as the ability of an organism to survive and reproduce in a given environment).

- Figure 2: Normalisation should be to OD and volume rather than to assumed protein amounts.

We have now normalized the data (Figure 3) to OD₆₀₀ units.

- Figure 3 and Figure S1 the authors should consider to include quantification of the cell lengths in addition to the stalk lengths as the increase in cell size is also indicative of phosphate starvation.

A quantification of the cell lengths and widths has now been included (new Supplementary Figures 1 and 3).

- Figure 4, Figure S2, Figure S3A: The scale for the growth curves should be in log scale.

A presentation of the growth data in logarithmic scale is the method of choice to determine growth rates, but it largely obscures the different time points of entry into stationary phase and the different final optical densities of the cultures. However, these parameters provide important insights into the physiological states of the mutant strains. Therefore, we would prefer to keep the linear scale.

- Figure 4: It would be helpful to have the PitA minus and plus plotted on the same axis to more clearly compare the conditions.

The panels looked very convoluted when all six curves were included in the same graph. Therefore, we would prefer to keep the current format.

- Figure 7: In panel A the d_pstS+PitA vs WT appears to be mislabelled, should it not be d_pstS vs WT+PitA? This figure is very difficult to make sense of and perhaps panels A and B could be more clearly labelled.

Thank you for pointing out this error. We have now corrected the labeling. Moreover, we have added labels to panels A and B that describe the conditions analyzed in each of the two Venn diagrams to make the figure (now **Figure 8**) easier to understand.

- Table 1: What is the source of the intracellular phosphate levels? Is this experimentally validated, an assumption or from literature? The authors should make this clearer and include references if needed.

We now provide the source of the information presented in Table 1 in the form of footnotes.

Reviewer #2

In this report by Billini et al, the authors investigate distinct pathways regulated by intracellular and extracellular phosphate availability in *C. crescentus*. The canonical phosphate uptake system in *C. crescentus*, consisting of PstSCAB, regulates the activity of the PhoRB two component system to induce expression of phosphate-regulated genes. A clear phenotype of phosphate limitation, due to low exogenous phosphate levels or deletion of the *pstS* phosphate, is the extension of the polar stalk. While the Pst system monitors environmental phosphate concentrations, other bacteria appear to have systems to simultaneously assess cytoplasmic phosphate levels. Unlike other organisms which have multiple phosphate uptake systems, *Caulobacter* appears to exclusively use the Pst transporter. This makes it a good model system to investigate the distinct roles of extracellular and cytoplasmic phosphate in regulating gene transcription. The authors exogenously expressed the PitA transporter from *E. coli* to enable phosphate uptake in a *pstS* mutant. This strain would have high intracellular phosphate, but would be unable to activate PhoB. Using a series of pair-wise RNAseq experiments, the authors identified a set of genes regulated by cytoplasmic phosphate independent of PhoB. They also identified genes specifically under the control of PhoB. Interestingly, these were non-overlapping sets of genes. Most surprisingly, they found that stalk elongation, the most characteristic phenotype of phosphate limitation, is not regulated by PhoB.

Overall, this paper uses a very nice genetic system to tease apart the various mechanisms underlying the *Caulobacter* response to changes in phosphate availability. The conclusions are consistent with the data presented and I have no major concerns.

I do have a couple of suggestions that may help strengthen the paper further.

1. Since *Caulobacter* does not have a second low-affinity phosphate transporter, it would seem that PstS is always responsible for phosphate import. Yet only when levels drop dramatically does it induce PhoB activation. If so, would the "cytoplasmic" phosphate sensor be triggered at concentrations higher than PhoB. What would happen if cells were grown in M2G (or HIGG) with varying concentrations of phosphate. Your data predict that there should be an order to which genes are

turned on first as phosphate levels drop. That could be tested by qRT-PCR using candidate genes from the RNAseq experiments.

We agree that it is interesting to determine experimentally in which order genes are turned on in response to different degrees of phosphate starvation. However, the PstSCAB system is a high-affinity transporter with an apparent K_D value for phosphate of 3.2 μM (see Figure 3), so that concentrations in the range of 0-5 μM would be required to establish conditions in which the phosphate uptake rate falls below the maximum value. It would not be possible to cultivate cells in these conditions for long enough to allow an equilibration of the system, because the constant uptake of phosphate would lead to a rapid fractional change in the phosphate concentration, making it impossible to correlate gene expression with fixed phosphate concentrations. Therefore, the type of experiment proposed by the reviewer is difficult to realize. We would also like to point out that the two gene sets regulated only under severe or moderate phosphate starvation, respectively, were determined using mutant strains that differ in the presence of PhoB, so that their composition may, to a certain degree, be influenced by the strain backgrounds.

Given its high affinity for phosphate, the PstSCAB transporter will work at maximal rate, and thus ensure a high cytoplasmic phosphate level, until essentially all extracellular phosphate is consumed. When its transport rate drops, PhoB is activated and induces an array of genes whose products mobilize phosphate from alternative extra- and intracellular phosphate sources (such as nucleotides, polyphosphates and phospholipids). This process likely ensures that cytoplasmic phosphate homeostasis is maintained over a prolonged period of time. Only when these alternative sources are consumed, the cytoplasmic phosphate level will drop and thus induce a change in the expression of cytoplasmic phosphate-regulated genes, preparing cells for growth and persistence under global phosphate deprivation.

To verify this sequence of events, we generate a series of new *lacZ* reporter constructs to monitor changes in the expression of various PhoB- and cytoplasmic phosphate-regulated genes after a shift of cells to phosphate-free ($\text{M2G}^{-\text{P}}$) medium (new Figure 9). As expected, we observed that PhoB-dependent promoters responded immediately to phosphate deprivation, consistent with the finding that the PstSCAB-PhoRB pathway responds exclusively to the extracellular stimuli. Cytoplasmic phosphate-regulated genes, by contrast, were upregulated asynchronously at later time points, likely responding to different degrees of cytoplasmic phosphate deprivation. These results are now described as a new final paragraph in the Results section.

A detailed analysis of the stepwise induction of cytoplasmic phosphate-regulated genes would require in-depth time-resolved global transcriptome studies that go beyond the scope of the present study.

2. Given the GO terms for genes regulated by the core vs severe phosphate limitation, you might use metabolomics to show that the predicted metabolic pathways are differentially regulated under these different phosphate regimes.

We suggest that the regulation of genes responding to cytoplasmic phosphate limitation may be achieved, at least in part, indirectly through changes in the accumulation of phosphate-containing metabolites. It would therefore be informative to perform time-resolved metabolomics studies to follow the reorganization of cellular metabolism over time and correlate the changes observed changes in the gene expression patterns. However, such studies would be time- and cost-intensive and are thus beyond the scope of the present study.

Reviewer #3

Billini et al. provide a detailed characterization of the physiological and transcriptional response of *C. crescentus* to changing phosphate levels. This is important, given the crucial role that phosphorus and phosphate availability plays in various essential cellular processes that support growth and proliferation. Using targeted deletion mutants and the heterologous expression of an *E. coli* phosphate transporter, they are able to disentangle contributions from environmental and cytoplasmic phosphate levels. With RNA sequencing experiments and downstream analyses comparing different strains, they show that a specific response is mounted to altered cytoplasmic phosphate levels, independent of the PhoR/PhoB two-component system and the PstSCAB transport system. The study is the first to identify the specific sensing and responding to alterations in cytoplasmic phosphate level in *C. crescentus*. In general, the study is well written and the results, figures, and discussion are clearly presented and interesting. The findings likely also have broader relevance, given that all bacteria face the challenge of modulating their growth and proliferation to current nutrient availabilities (which includes phosphate). Below, I have listed some (minor) aspects that I would like to see resolved.

Main comments

-The first section of the manuscript that focuses on the characterization (stalk elongation + growth) of the *phoB* deletion mutant appears a bit out of place and not in line with the second section. I understand that the goal is to set the stage somewhat for later experiments but wonder if these results cannot simply be incorporated there.

The first section indeed sets the stage for the subsequent analyses by determining in detail the phenotypes of the strains used in this study under all relevant conditions. In particular, the general growth defect of the Δ *phoB* mutant in phosphate-limiting conditions led us to doubt the direct involvement of PhoB in stalk elongation and to search for an alternative regulatory mechanism. We find it difficult to present these data later in the manuscript without disrupting the flow of arguments, especially because other reviewers requested new experiments related to this section that required the incorporation of additional explanations. Therefore, we would like to retain the original structure of the Results section. However, we tried to simplify the description of the results and emphasize the relevance of the finding to the analyses described in the later parts of the manuscript without further increasing the overall length of the text.

-The morphological characterizations currently focus solely on stalk lengths, while other factors (e.g., cell width and length) also appear to be affected. In addition, based on the transcriptional response to altered cytoplasmic phosphate levels, can the authors speculate on any of the molecular mechanisms that underlie cell morphological alterations in conditions where phosphate is limiting?

We have now included a quantification of cell lengths and widths for all strains and conditions (new Supplementary Figures 1 and 3).

At this stage, it is difficult to tell how that changes in gene expression observed in phosphate-limiting conditions lead to the characteristic morphological changes of *C. crescentus*. As mentioned in the Discussion, previous work by de Young *et al* (2020, *J. Bacteriol.*) showed that stalk elongation is abolished by mutations in the phosphomannose isomerase ManA (CCNA_03732) that reduce the enzymatic activity of the enzyme and thus shift the equilibrium between fructose-6-phosphate and mannose-6-phosphate. This finding indicates that phosphosugar metabolism is critically involved in the morphological transition induced by phosphate starvation. However, given that phosphorylated sugars play critical roles in a large variety of biosynthetic and regulatory pathways and fructose-6-

phosphate is a central metabolite whose levels may affect the flux through multiple pathways, the precise connection between ManA and stalk elongation remains unclear.

Overall, phosphate-starved cells appear to be locked at the stalked-cell stage, in which cells typically elongate the stalk and cell body. The continued growth of the stalk and cell bodies may thus be an indirect consequence of this developmental block, which may be induced by a combination of transcriptional and metabolic changes. Since all mechanistic models connecting phosphate-regulated gene expression and cell morphology are pure speculation, we would prefer not to expand on this point in the Discussion.

-Despite that there is limited overlap between the identified PhoB regulon and the core cytoplasmic phosphate regulon, the authors still mention an interplay between extracellular and intracellular phosphate sensing? To me, it is currently unclear how this interplay would be established (and what would be the importance of this).

The overlap between the PhoB regulon and the cytoplasmic phosphate regulon observed in our analysis is indeed quite small (8 genes). It is conceivable that some genes may be subject to dual regulation, mediated e.g. by the interaction of their promoter regions with multiple regulators or indirect effects of one regulon on the expression of genes in the other regulon. Such as crosstalk could help to fine-tune phosphate uptake and mobilization with metabolic readjustments during up- or downshifts in extracellular phosphate availability or short-term fluctuations in cytoplasmic or environmental phosphate availability.

-Line 328-331: how do the authors explain the differences in the phoB regulon that they identified and the one that was previously characterized. Do differences in experimental approach play a role? If so, it would be nice if the authors could elaborate on this to make this more clear.

The differences may indeed be explained by the different experimental approaches used. Lubin *et al* (2015) used a combination of microarray and ChIP-seq analysis to define the direct targets of PhoB, whereas the PhoB regulon defined in our study comprises all genes that are regulated exclusively by PhoB and not by the intracellular phosphate level, including both direct and indirect targets. We have made this clearer now in the Results section.

Notably, several of the genes identified by Lubin *et al* appear to be regulated in a PhoB-dependent manner in some of the comparisons we made but did not fulfill all criteria set to be included in the list of genes representing the final PhoB regulon (see Supplementary Data 2). These genes typically show relatively small changes in expression in the different conditions analyzed, so that their differential presence in the final PhoB regulons established by Lubin *et al* and this study, respectively, may be explained by different analytical pipelines and different thresholds for the fold change and statistical significance of these changes used to define the final regulons. We have now mentioned this possibility in the Results part.

-Given that PhoB typically functions as a transcriptional activator (or at least, that is how its presented in Figure 1), how do the authors explain the subset of genes that is downregulated when PhoB is ON (Figure 7)?

Out of the 47 genes in the PhoB regulon, four genes appear to be downregulated upon PhoB activation. Given that the PhoB regulon established in this study includes both direct and indirect targets of PhoB, this phenomenon is most likely explained by indirect effects mediated by the upregulation of direct PhoB targets. For instance, the regulon includes a MarR-type transcriptional regulator that could act as a repressor. Alternatively, the PhoB-dependent repression of genes could be induced by metabolic changes resulting from the activation of the PhoB regulon.

Minor comments

-Line 41: “the” appears to be missing in between “In this model,” and “phosphate-loaded state...”

Corrected.

Reviewer #4

Overall, the manuscript is well-written and addresses an interesting (and previously outstanding) question in the field (mostly *Caulobacter* folks). That is, how does *Caulobacter* regulate phosphate uptake (extracellularly and intracellularly) and how does this relate to cell shape and morphology? In terms of methodology, it appears that the authors have performed detailed and robust analyses, which lead to strong, novel findings.

I do worry about the utility of these findings across a broader scope.

It is not clear to us why reviewer #4 doubts the general significance of our results. We use *C. crescentus* as a model system to study phosphate-dependent regulation in bacteria, because its morphology offers a convenient readout of its physiological state. However, there is no reason to believe that the clear distinction between extracellular (PhoB-mediated) and intracellular phosphate sensing identified in our study cannot apply to other bacteria. Actually, we propose that the biphasic response to extracellular and intracellular phosphate depletion described in our paper is a general phenomenon in bacteria, because it does not make sense for cells to launch a full-fledged phosphate starvation response once environmental phosphate becomes limiting as long as they can still mobilize inorganic phosphate from intracellular storage compounds or alternative extracellular phosphorus donors (such as organic phosphates or phosphonates).

But I think that the work should be accepted for publication. The authors should, however, revise their Results section to either (a) omit much of the discussion points or (b) include the important parts from the Discussion section into the Results section. In its current form, the Results section reads more like a Results + Discussion section. Similarly, the Introduction section is a bit lengthy and could benefit from some concision.

We are aware that several aspects of the discussion are already included in the Results section. However, we realized that the train of thoughts and the rationale of the experiments are easier to grasp if some of our conclusions, in particular those on less central findings, are already drawn during the presentation of the data. As for the Introduction section, we find it important to present both the *E. coli* and the *C. crescentus* system and provide all information that is required to understand the experiments performed and the relevance of the study, in particular for readers who are not perfectly familiar with the field of phosphate regulation or the *Caulobacter* system. For these reasons, we would prefer not to reduce the length of the Introduction or the organization of the Results section.

REVIEWERS' COMMENTS:

Reviewer #1 (Remarks to the Author):

The authors have thoroughly revised their manuscript based on the reviewers' comments and suggestions. I have no further comments that need to be addressed.

Reviewer #2 (Remarks to the Author):

The revised manuscript by Billini et al adequately addresses my concerns and appears to largely address the concerns of my fellow reviewers.

The new data using the LacZ fusions (Figure 9) demonstrates the differential timing of expression for PhoB-dependent and cytoplasmic-Pi-dependent genes upon phosphate depletion.

One point that requires addressing is an apparent swap of data in the new Supplemental Figure S12. The stalk length quantification data in panel B does not match the images in panel A. It looks as if the data for the +/- P_{xyl}-pitA were swapped.

Reviewer #3 (Remarks to the Author):

I would like to thank the authors for carefully addressing all (minor) concerns I had. I have no remaining remarks with regard to the manuscript.